# Quantifying Representation Reliability in Self-Supervised Learning Models

**Young-Jin Park**[1]     **Hao Wang**[2]     **Shervin Ardeshir**     **Navid Azizan**[1]

[1]Massachusetts Institute of Technology
[2]MIT-IBM Watson AI Lab

## Abstract

Self-supervised learning models extract general-purpose representations from data. Quantifying the reliability of these representations is crucial, as many downstream models rely on them as input for their own tasks. To this end, we introduce a formal definition of *representation reliability*: the representation for a given test point is considered to be reliable if the downstream models built on top of that representation can consistently generate accurate predictions for that test point. However, accessing downstream data to quantify the representation reliability is often infeasible or restricted due to privacy concerns. We propose an ensemble-based method for estimating the representation reliability without knowing the downstream tasks a priori. Our method is based on the concept of *neighborhood consistency* across distinct pre-trained representation spaces. The key insight is to find shared neighboring points as anchors to align these representation spaces before comparing them. We demonstrate through comprehensive numerical experiments that our method effectively captures the representation reliability with a high degree of correlation, achieving robust and favorable performance compared with baseline methods. The code is available at https://github.com/azizanlab/repreli.

## 1 INTRODUCTION

Self-supervised learning (SSL) has opened the door to the development of general-purpose embedding functions, often referred to as *foundation models*, that can be used or fine-tuned for various downstream tasks [Jaiswal et al., 2020, Jing and Tian, 2020]. These embedding functions are pre-trained on large corpora of different data modalities, span-

ning visual [Chen et al., 2020a], textual [Brown et al., 2020], audio [Al-Tahan and Mohsenzadeh, 2021], and their combinations [Radford et al., 2021, Morgado et al., 2021], aimed at being general-purpose and agnostic to the downstream tasks they may be utilized for. For instance, the recent surge in large pre-trained models such as CLIP [Radford et al., 2021] and ChatGPT [OpenAI, 2022] has resulted in the development of many prompt-based or dialogue-based downstream use cases, none of which are known a priori when the pre-trained model is being deployed.

Embedding functions learned through SSL may produce unreliable outputs. For example, large language models can generate factually inaccurate information with a high level of confidence [Bommasani et al., 2021, Tran et al., 2022]. With the increasing use of SSL to generate textual, visual, and audio content, unreliable embedding functions could have significant implications. Furthermore, given that these embedding functions are frequently employed as frozen backbones for various downstream use cases, adding more labeled downstream data may not improve the performance if the initial representation is unreliable. Therefore, having notion(s) of **reliability/uncertainty** for such pre-trained models, alongside their abstract representations, would be a key enabler for their reliable deployment, especially in safety-critical settings.

In this paper, we introduce a formal definition of representation reliability based on its impact on downstream tasks. Our definition pertains to a representation of a given test point produced by an embedding function. If a variety of downstream tasks that build upon this representation consistently yield accurate results for the test point, we consider this representation reliable. Existing uncertainty quantification (UQ) frameworks mostly focus on the supervised learning setting, where they rely on the consistency of predictions across various predictive models. We provide a counter-example showing that they cannot be directly applied to our setting, as representations lack a ground truth for comparison. In other words, inconsistent predictions often indicate that the predictions are not reliable, but inconsistent repre-

sentations do not necessarily imply that the representations are unreliable. Hence, it is critical to align representation spaces in such a way that corresponding regions have similar semantic meanings before comparing them.

We propose an ensemble-based approach for estimating representation reliability without knowing downstream tasks a priori. We prove that a test point has a reliable representation if it has a reliable neighbor which remains consistently close to the test point, across multiple representation spaces generated by different embedding functions. Based on this theoretical insight, we select a set of embedding functions and reference data (e.g., data used for training the embedding functions). We then compute the number of consistent neighboring points among the reference data to estimate the representation reliability. The underlying reasoning is that a test point with more consistent neighbors is more likely to have a reliable and consistent neighbor. This reliable and consistent neighbor can be used to align different representation spaces that are generated by different embedding functions.

We conduct extensive numerical experiments to validate our approach and compare it with baselines, including state-of-the-art out-of-distribution (OOD) detection measures and UQ in supervised learning. Our approach consistently captures the representation reliability in all different applications. These applications range from predicting the performance of embedding functions when adapting them to in-distribution or out-of-distribution downstream tasks to ranking the reliability of different embedding functions. While the baselines may occasionally surpass our approach, their performance fluctuates significantly across different settings and can even become negative, posing a risk when used to assess reliability in safety-critical settings.

In summary, our main contributions are:

- We present a formal definition of representation reliability, which quantifies how well an abstract representation can be used across various downstream tasks. To the best of our knowledge, this is the first comprehensive study to investigate uncertainty in representation space.

- We provide a counter-example, showing that existing UQ tools in supervised learning cannot be directly applied to estimate the representation reliability.

- We prove a theorem stating that a reliable and consistent neighbor of a test point can serve as an anchor point for aligning different representation spaces and assuring the representation reliability.

- Based on our theoretical findings, we introduce an ensemble-based approach that uses neighborhood consistency to estimate the representation reliability.

- We conduct comprehensive numerical experiments, showing that our approach consistently captures the representation reliability whereas the baselines could not.

**Broad Impact and Implication.** Our work introduces a way to quantify the reliability of pre-trained models prior to their deployment in specific downstream tasks, which has several implications. Imagine a practitioner has access to multiple pre-trained models that have been trained using distinct learning paradigms, data, or architectures. Our method helps compare and rank these models based on their reliability scores, enabling the practitioner to select the model with the highest reliability score. This is particularly valuable when transmitting downstream training data is challenging due to privacy concerns, or when the downstream tasks shift over time. Similarly, in cases where a pre-trained model yields incorrect (or harmful) decisions for specific individuals, our method can help explain whether the issue stems from the abstract representation or the projection heads added to the pre-trained model. We hope our effort can push the frontiers of self-supervised learning towards more responsible and reliable deployment, encouraging further research to ensure the transferability of knowledge acquired during pre-training across diverse tasks and domains.

## RELATED WORKS

**Uncertainty Quantification in Supervised Learning.** Existing work on UQ mostly focused on supervised learning settings. For example, Bayesian inference quantifies uncertainty by placing a prior distribution over model parameters, updating this prior distribution with observed data to obtain a posterior distribution, and examining the inconsistency of predictions derived from the posterior distribution [Neal, 1996, MacKay, 1992, Kendall and Gal, 2017, Depeweg et al., 2018]. Since the posterior distribution may not have an analytical form, many approximating approaches have been introduced, including Monte Carlo dropout [Gal and Ghahramani, 2016], deep ensembles [Osband et al., 2018, Lakshminarayanan et al., 2017, Wen et al., 2020], and Laplace approximation [Daxberger et al., 2021, Sharma et al., 2021]. In this paper, we focus on quantifying the uncertainty of representations and prove that standard supervised-learning frameworks cannot be directly applied to investigate representation uncertainty (see Section 3.2 for more details).

**Novelty Detection and Representation Reliability.** Self-supervised learning is increasingly used for novelty/OOD detection. These approaches train self-supervised models and then compute an OOD score for a new test point based on its distance from the training data in the representation space [Lee et al., 2018, van Amersfoort et al., 2020, Tack et al., 2020, Mirzaei et al., 2022]. However, OOD detection and our representation reliability are different concepts. The former identifies whether a test point belongs to the same distribution as the (pre-)training data, while the latter evaluates the possibility that a test point can receive accurate predictions when the self-supervised learning model is adapted to various downstream tasks (see Section 3.1 for

more details).

To compare with this line of work, we conduct comprehensive numerical experiments (Section 4). The results suggest that our approach more robustly captures the representation reliability compared with state-of-the-art OOD detection measures and the empirical metrics proposed in Ardeshir and Azizan [2022]. Finally, our representation reliability extends the notion of probe as in HaoChen et al. [2021] to multiple downstream tasks. We introduce an algorithm for estimating the representation reliability without prior knowledge of the specific downstream tasks.

**Uncertainty-Aware Representation Learning.** There is a growing body of research aimed at training robust self-supervised models that map input points to a distribution in the representation space, rather than to a single point [Vilnis and McCallum, 2015, Neelakantan et al., 2014, Karaletsos et al., 2016, Bojchevski and Günnemann, 2018, Oh et al., 2018, Chen et al., 2020a, Wu and Goodman, 2020, Zhang et al., 2021, Almecija et al., 2022]. They rely on special neural network architectures and/or introduce alternative training schemes. For example, the approach by Zhang et al. [2021] requires an additional output (i.e., a temperature parameter) and the approach by Oh et al. [2018] requires the network to output means and variances of a mixture of Gaussian distributions. In contrast, we avoid making any assumptions about the training process of the embedding functions, while only needing black-box access to them. Furthermore, we provide a theoretical analysis of our method and explore the impact of the representation reliability on the performance of downstream tasks. We provide a more in-depth discussion about related works in Appendix D.

## 2 BACKGROUND: UNCERTAINTY QUANTIFICATION IN SUPERVISED LEARNING

We recall a Bayesian-inference view of epistemic uncertainty in supervised learning. Consider predictive models, where each model (e.g., a neural network) $f$ outputs a predictive probability $p(y|\boldsymbol{x}, f)$ for an input variable $\boldsymbol{x}$. In the Bayesian framework, a prior probability distribution $p(f)$ is first introduced and a posterior distribution is learned given a training dataset $\mathcal{D}$: $p(f|\mathcal{D}) \propto p(\mathcal{D}|f) \cdot p(f)$. For a new test point $\boldsymbol{x}^*$, its posterior predictive distribution is obtained by averaging the predictive probabilities over models:

$$p(\hat{y}^* \mid \boldsymbol{x}^*, \mathcal{D}) = \int p(\hat{y}^* \mid \boldsymbol{x}^*, f)p(f \mid \mathcal{D})\mathrm{d}f. \quad (1)$$

Since the posterior distribution does not have an analytical expression for complex neural networks, Monte Carlo approaches could be used to approximate the integral in Equation (1). One of the most prominent approaches is deep ensembles [Lakshminarayanan et al., 2017], in which a class of neural networks $\mathcal{F}$ are trained with $M$ different random initialization of the learnable parameters $\{\theta_i\}_{i=1}^M$. The posterior predictive distribution is then approximated by:

$$p(\hat{y}^* \mid \boldsymbol{x}^*, \mathcal{D}) \approx \frac{1}{M} \sum_{i=1}^M \left[ \delta(\hat{y}^* - f_{\theta_i}(\boldsymbol{x}^*)) \right]. \quad (2)$$

Finally, the uncertainty can be assessed by the *inconsistency* of the output across sampled functions, which can be quantified by metrics such as variance[1] (for regression tasks) or entropy (for classification tasks). For example, in regression tasks, the uncertainty can be estimated by

$$\mathsf{Unc}\,(\boldsymbol{x}^* \mid \mathcal{D}) = \mathsf{Var}\,(\hat{y}^* \mid \boldsymbol{x}^*, \mathcal{D}) \quad (3)$$
$$\approx \mathsf{Var}_{i\sim[M]}\,(f_{\theta_i}(\boldsymbol{x}^*)) = \frac{1}{M^2} \sum_{i<j}\|f_{\theta_i}(\boldsymbol{x}^*) - f_{\theta_j}(\boldsymbol{x}^*)\|_2^2,$$

where $\mathsf{Var}_{i\sim[M]}\,(\cdot)$ is the population variance over index $i \in \{1, \cdots, M\}$. When different models disagree on their predictions, it suggests a high level of uncertainty. Conversely, if multiple predictive models give similar outputs, the prediction can be considered certain. However, this principle does not hold in self-supervised learning, as representations do not have a ground truth and different pre-trained models can carry different semantic meanings.

## 3 QUANTIFYING REPRESENTATION RELIABILITY

In this section, we introduce a framework for assessing the reliability of representations assigned by pre-trained models. We discuss limitations of existing UQ frameworks in supervised learning when applied to representation spaces. Then we present an ensemble-based method that examines the consistency of neighboring points in the representation space. Our method effectively captures the representation reliability without the need for a priori knowledge of downstream tasks.

### 3.1 REPRESENTATION RELIABILITY

We introduce a formal definition of *representation reliability* by examining its impact on various downstream tasks. Intuitively, a reliable representation assigned by a pre-trained model should consistently yield accurate outcomes when the model is adapted to these downstream tasks.

We introduce some notations that will be used in our definition. We define $\mathcal{H}$ as a class of embedding functions trained by a self-supervised learning algorithm (e.g., SimCLR [Chen et al., 2020a]) using a pre-trained dataset (e.g.,

---

[1]For a random vector, its variance is defined as the trace of its covariance matrix.

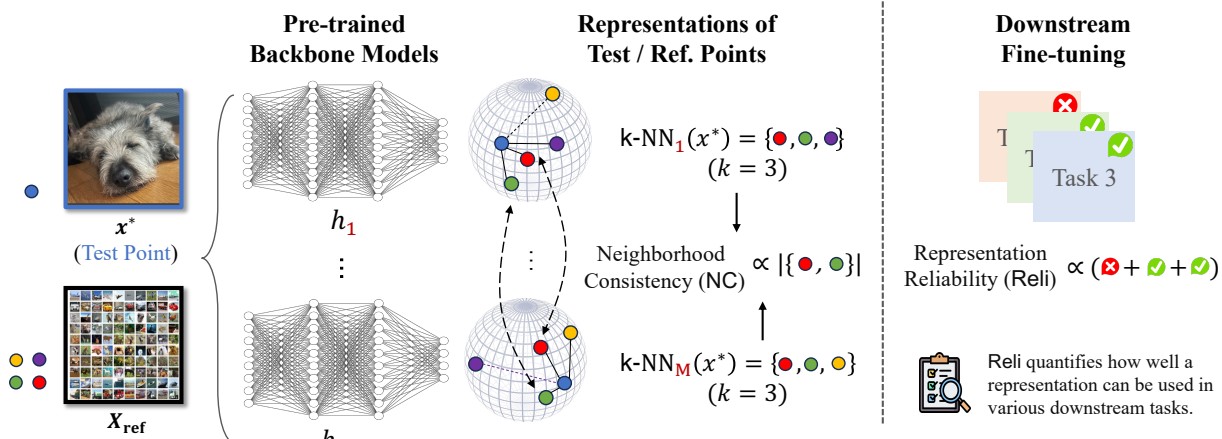

Figure 1: Illustration of *representation reliability* (Reli) and *neighborhood consistency* (NC). For a test point $x^*$ and a class of pre-trained backbone models $\mathcal{H} = \{h_1, \cdots, h_M\}$, the representation reliability is defined as the average performance of downstream models when using the representations of $x^*$ provided by the backbones in $\mathcal{H}$. **Our** NC **estimates** Reli **without requiring any prior knowledge of the downstream tasks.** It operates by measuring the number of consistent neighbors of $x^*$ among reference points across different representation spaces.

ImageNet [Deng et al., 2009]). Each embedding function $h : \mathcal{X} \to \mathcal{Z}$ maps a data point (e.g., an image) to an abstract representation. Here the representation space $\mathcal{Z}$ can be either a real $d$-space $\mathbb{R}^d$ or a unit hyper-sphere $\mathcal{S}^{d-1}$.

We consider a collection of downstream tasks (e.g., classifications), denoted as $\mathcal{T}$. Each task is associated with a set of downstream heads (i.e., additional layers added on top of the embedding function) and a population risk function that assesses the performance of each downstream head. For each task $t$, we optimize the risk function to obtain an optimal head $g_{h,t}$. This way, we can eliminate the impact of downstream training processes on our definition. The representation reliability of a new test point $x^*$ is measured by the (average) performance of these downstream models on $x^*$. A formal definition is provided below.

**Definition 1.** Let $\mathcal{T}$ be a collection of downstream tasks. For each task $t \in \mathcal{T}$, we take an embedding function $h$ (uniformly at random) from $\mathcal{H}$ and find an optimal downstream head $g_{h,t}$ based on $h$. We define the *representation reliability* for a test point $x^* \in \mathcal{X}$ as:

$$\mathsf{Reli}(x^*; \mathcal{H}, \mathcal{T}) \triangleq \frac{1}{|\mathcal{T}|} \sum_{t \in \mathcal{T}} \mathsf{Perf}_t\left(g_{h,t} \circ h,\ x^*\right) \quad (4)$$

where $\mathsf{Perf}_t(\cdot)$ measures the performance of the predictive model $g_{h,t} \circ h$ on $x^*$ for task $t$.

For classification downstream tasks with $C$ classes, an example of $\mathsf{Perf}_t(g_{h,t} \circ h,\ x^*)$ is the negative Brier score, defined as:

$$-\sum_{c=1}^{C} \left( \frac{1}{|\mathcal{H}|} \sum_{h \in \mathcal{H}} g_{h,t} \circ h(x^*)_{[c]} - y^*_{t[c]} \right)^2.$$

Here $y^*_t \in \{0, 1\}^C$ represents the label of $x^*$ on task $t$. For additional examples of $\mathsf{Perf}_t(\cdot)$, please refer to Appendix B.

The above definition assumes that the set of downstream tasks (and ground-truth labels) are accessible. In practice, this may not always be the case (see Broad Impact and Implication in Section 1 for examples). Next, we discuss how to estimate the representation reliability based on the properties of the representation itself, without prior knowledge of the downstream tasks.

## 3.2 FIRST ATTEMPT: REPRESENTATION CONSISTENCY

Our first attempt is directly applying standard supervised-learning techniques (see Section 2) to estimate the representation reliability. Recall that if multiple predictive models give different predictions for the same test point, then it is likely that their predictions are uncertain. One may wonder whether the same idea can be applied to estimate the representation reliability. We present a negative result, showing that even if different embedding functions produce completely different representations, their downstream predictions can still be consistent. Below, we provide an (informal) theorem and defer a more rigorous statement along with its proof to Appendix A.1.

**Theorem 1.** *For any constant $A$ and a test point $x^*$, there exist embedding functions $h_1, \cdots, h_M \in \mathcal{H}$ such that $\mathsf{Var}_{i \sim [M]}(h_i(x^*)) \geq A$ but $\mathsf{Var}_{i \sim [M]}(g_{i,t} \circ h_i(x^*)) = 0$ for any downstream task $t$. Here $g_{i,t}$ is an optimal downstream head for $h_i$ under task $t$.*

The key insight behind our proof is that an input point's

representation is not unique (e.g., rotated spaces are in fact equivalent). In other words, even if different embedding functions assign distinct representations to the same test point, downstream heads built on these embedding functions can also vary, ultimately leading to similar predictions.

## 3.3 PROPOSED FRAMEWORK: NEIGHBORHOOD CONSISTENCY

To address the aforementioned issue, we propose the idea of using an "anchor" point to align different representation spaces. The anchor point serves as a bridge that transforms different representation spaces into the same space. The graphical visualization of this idea is illustrated in Figure 1 and Figure 3. We formalize this intuition more rigorously in the following theorem. It states that if a test point has a (reliable) consistent neighboring point across all representation spaces, then uncertainty of its downstream predictions are bounded above.

**Theorem 2.** *For a test point $x^*$, suppose that there exists a **consistent** neighbor $x^r$ across all embedding functions $\mathcal{H} = \{h_1, \cdots, h_M\}$, satisfying*

$$\|h_i(x^r) - h_i(x^*)\|_2 \leq \epsilon_{nb} , \ \forall i \in [M]. \quad (5)$$

*Suppose the downstream heads $g_{i,t}$ are Lipschitz continuous. Then, for any downstream task $t$, the variance of downstream prediction at $x^*$ is bounded above by:*

$$\mathsf{Var}_{i \sim [M]} \left( g_{i,t} \circ h_i(x^*) \right) \leq (\sqrt{2} L_t \epsilon_{nb} + \sigma_{r,t})^2. \quad (6)$$

*where $\sigma_{r,t}^2 = \mathsf{Var}_{i \sim [M]} \left( g_{i,t} \circ h_i(x^r) \right)$ is the reliability of $x^r$ as measured by variance, the function $g_{i,t}$ is the optimal downstream head for task $t$ built upon $h_i$, whose Lipschitz constant is $L_{i,t}$, and $L_t = \max_i L_{i,t}$.[2]*

See Appendix A.2 for proof. The key takeaway is that if we can find a reference point consistently close to the test point, it can serve as an anchor point that helps align representation spaces with distinct semantic meanings. Plus, the reliability of the reference point along with its relative distance to the test point ensures a lower bound on the representation reliability of the test point.

In practice, identifying a reliable reference point is challenging without prior knowledge of the downstream tasks. Instead of searching for a single point as the anchor, we draw a set of reference points, denoted as $X_{\text{ref}} = \{x^{(l)}\}_{l=1}^n$. We then compute the number of consistent neighboring points within $X_{\text{ref}}$ and use it to estimate the representation reliability. The rationale behind this is that a test point with more consistent neighbors is more likely to have a reliable and consistent neighbor.

---

[2]When spectral normalized weights are employed with 1-Lipschitz continuous activation functions (e.g., identity, ReLU, sigmoid, or softmax), the constant $L_t = 1$.

**Our Algorithm.** Given an ensemble of embedding functions $h_1, \cdots, h_M$ and $X_{\text{ref}}$, we define the **Neighborhood Consistency (NC)** of a test point $x^*$ as:

$$\mathsf{NC}_k(x^*) = \frac{1}{M^2} \sum_{i < j} \mathsf{Sim} \left( k\text{-NN}_i(x^*), \ k\text{-NN}_j(x^*) \right)$$

$$(7)$$

where $k\text{-NN}_i(x^*)$ is the index set of $k$-nearest neighbors of $h_i(x^*)$ among $\{h_i(x) \mid x \in X_{\text{ref}}\}$ and $\mathsf{Sim}(\cdot, \cdot)$ is a measure of similarity between sets (e.g., *Jaccard Similarity*).

**Remark 1.** The parameter $k$ involves a trade-off between two factors: choosing a neighborhood closer to the test point and increasing the chance of incorporating a more reliable neighbor. As shown in Theorem 2, the reliability of a test point's representation hinges on whether it has a nearby reliable neighbor. Hence, a large $k$ may result in selecting a reliable neighbor, but it could be far from the test point. On the other hand, since the upper bound in Theorem 2 holds using any one of the neighbors as the anchor point, the reliability of the test point is bounded using the neighbor with the smallest $\sigma_{r,t}$. Consequently, a small $k$ may compromise the reliability of the selected neighboring point. We present an empirical study about this trade-off in Section 4.5.

Finally, our NC requires a set of embedding functions for computing consistent neighbors. We extend this algorithm to evaluate the reliability of a single embedding function in Section 4.4.

## 4 NUMERICAL EXPERIMENTS

### 4.1 EXPERIMENT SETUP

**Embedding Function.** We use SimCLR [Chen et al., 2020a], Bootstrap Your Own Latent (BYOL) [Grill et al., 2020], and Momentum Contrast (MoCo) [He et al., 2020] to pre-train ResNet-18 and ResNet-50 models [He et al., 2016] as the embedding functions, using a single NVIDIA V100 GPU. During the pre-training stage, we do not use any class label information. To construct the ensemble, we train a total of $M = 10$ embedding functions using each pre-training algorithm and dataset with different random seed, following Lakshminarayanan et al. [2017]. The impact of $M$ to our experiment is studied in Section 4.5.

**Baselines.** We compare our proposed method — $\mathsf{NC}_k$ in Equation (7) — against the following widely-recognized baseline methods:

- $\mathsf{Dist}_k$ [Tack et al., 2020, Mirzaei et al., 2022]: the average of the $k$ minimum distances from the test point to the reference data in the representation space. A lower value of $\mathsf{Dist}_k$ indicates higher reliability.

Table 1: Comparison of our neighborhood consistency ($NC_{100}$) with baseline methods in terms of their correlation with the representation reliability. We use Kendall's $\tau$ coefficient to measure this correlation. Downstream tasks are in-distribution tasks where the model is pre-trained and fine-tuned on the same dataset. Performance on these tasks is evaluated using either negative predictive entropy or negative Brier score. The highest and second-highest scores are highlighted in bold and underlined, respectively. As shown, our method consistently receives a favorable score compared with baselines.

| Pretraining Algorithms | Method | ResNet-18 | | | | ResNet-50 | | | |
| | | CIFAR-10 | | CIFAR-100 | | CIFAR-10 | | CIFAR-100 | |
| | | Entropy | Brier | Entropy | Brier | Entropy | Brier | Entropy | Brier |
|---|---|---|---|---|---|---|---|---|---|
| SimCLR | $NC_{100}$ | **0.3865** | **0.3262** | **0.3130** | **0.2590** | **0.3786** | **0.3206** | **0.3070** | **0.2516** |
| | $Dist_1$ | 0.3021 | 0.2544 | 0.0757 | 0.0635 | 0.2750 | 0.2376 | 0.1100 | 0.0919 |
| | Norm | 0.2907 | 0.2429 | 0.1198 | 0.0649 | 0.2789 | 0.2385 | 0.1311 | 0.0710 |
| | LL | 0.1797 | 0.1356 | -0.0989 | -0.1067 | 0.1700 | 0.1348 | -0.0831 | -0.0959 |
| | FV | -0.0476 | -0.0458 | -0.1891 | -0.1710 | -0.0402 | -0.0393 | -0.1984 | -0.1870 |
| BYOL | $NC_{100}$ | **0.2749** | **0.1826** | **0.2354** | **0.1753** | **0.3524** | **0.2131** | **0.3233** | **0.1733** |
| | $Dist_1$ | 0.0322 | 0.0385 | -0.1477 | -0.0709 | -0.1858 | -0.1442 | -0.3124 | -0.1598 |
| | Norm | 0.0725 | 0.0235 | 0.1659 | 0.1231 | 0.1990 | 0.1184 | 0.1990 | 0.1476 |
| | LL | -0.0539 | -0.0196 | -0.2637 | -0.1650 | -0.2542 | -0.1784 | -0.3777 | -0.2018 |
| | FV | 0.1439 | 0.1003 | -0.1727 | -0.1284 | 0.0676 | 0.0524 | -0.0213 | -0.0625 |
| MoCo | $NC_{100}$ | **0.4157** | **0.3653** | **0.3632** | **0.3128** | 0.3757 | 0.3236 | **0.2667** | **0.2225** |
| | $Dist_1$ | 0.3523 | 0.3093 | 0.2246 | 0.1925 | **0.3831** | **0.3397** | 0.2453 | 0.2183 |
| | Norm | 0.3238 | 0.2708 | 0.2186 | 0.1432 | 0.3798 | 0.3294 | 0.2172 | 0.1659 |
| | LL | 0.2201 | 0.1744 | -0.0334 | -0.0736 | 0.2518 | 0.2125 | -0.0156 | -0.0343 |
| | FV | 0.0519 | 0.0321 | -0.1420 | -0.1578 | 0.1228 | 0.1122 | -0.1081 | -0.1100 |

- Norm [Tack et al., 2020]: the $L_2$ norm of the representation $\|h(\boldsymbol{x}^*)\|_2$. A higher value of Norm indicates higher reliability.

- LL [Ardeshir and Azizan, 2022]: the log-likelihood of the Gaussian mixture model ($\mathcal{R}^d$) or the von Mises–Fisher [Banerjee et al., 2005] ($\mathcal{S}^{d-1}$) mixture model on the test point when fitted on the reference data. A higher value of LL indicates higher reliability.

- Feature Variance (FV): representation consistency measured by $\mathsf{Var}_{i \sim [M]}(h_i(\boldsymbol{x}^*))$, as described in Theorem 1. FV is extended from UQ in supervised learning. Specifically, the variance of neural networks' predicted scores is often used to measure epistemic uncertainty [Kendall and Gal, 2017, Lakshminarayanan et al., 2017, Ritter et al., 2018]. Here we apply this measure to examine latent representation spaces. A lower value of FV indicates higher reliability.

All baselines, except FV, are based on a single embedding function. For a fair comparison, we consider the (point-wise) ensemble average of each score over different embedding functions for $Dist_k$, Norm, and LL.

**Hyperparameters.** To reduce the computational cost, we randomly select $n = 5,000$ pre-training data as the reference dataset $\boldsymbol{X}_{\mathrm{ref}}$. We repeat our experiments 5 times with distinct random seeds for choosing reference data and report the average evaluation scores. Since the standard deviations of our scores are on average below 1%, we defer the experimental results with error bars to Appendix. We choose $k = 100$ for $NC_k$ and $k = 1$ for $Dist_k$ (see Section 4.5 for

an ablation study about the choice of $k$).

For our method and $Dist_k$, we test both cosine distance and Euclidean distance as options for distance metric. In a similar manner, LL and FV are evaluated using both unnormalized $h(\boldsymbol{x}) \in \mathbb{R}^d$ and normalized representations $h(\boldsymbol{x})/\|h(\boldsymbol{x})\|_2 \in \mathcal{S}^{d-1}$. A discussion on the implications of these selections is provided in Section 4.5. Given the space limits, the results using normalized representations are reported in the main manuscript, while those using unnormalized representations are detailed in Appendix C.1.

**Evaluation Protocol.** To evaluate the effectiveness of our method (i.e., $NC_{100}$) in capturing the representation reliability (Definition 1) and to compare it against baselines[3], we use Kendall's $\tau$ coefficient to measure the correlation between each method and the representation reliability (Reli). We compute the downstream performance (Perf) on each test point by using either 1) the negative predictive entropy, capturing downstream uncertainty; or 2) the negative Brier score, reflecting predictive accuracy. The computation details are provided in Appendix B.

We focus on downstream classification tasks. For each task, we freeze the pre-trained model and fine-tune the linear heads. To leverage the multi-class labels and minimize the influence from the downstream training processes, we break down each $C$-class classification into a set of one-vs-one (OVO) binary classification tasks. As a result, the total number of downstream tasks is $|\mathcal{T}| = C(C-1)/2$, with each

---

[3]We use the negative scores of $Dist_k$ and FV since their lower values indicate higher reliability.

data point being evaluated in $(C-1)$ tasks. Finally, we average the performance across all OVO tasks to compute the representation reliability.

## 4.2 MAIN RESULTS

### 4.2.1 Correlation to ID Downstream Performance

Recall that the representation reliability relies on downstream tasks. Here we focus on in-distribution (ID) tasks where embedding functions are pre-trained on `CIFAR-10` or `CIFAR-100` datasets [Krizhevsky et al., 2009] without using any labeling information. Subsequently, they are fine-tuned with a linear head on the same dataset.

Table 1 shows that our $NC_{100}$ demonstrates a higher correlation with the representation reliability compared with baselines. Notably, regardless with the choice of the pre-training configurations or downstream tasks, our method always has a positive correlation whereas baselines occasionally demonstrates very low or even negative correlation. Moreover, FV presents low correlation and this observation aligns with the theoretical results in Section 3.2.

We offer insight into why $NC_k$ has a more favorable performance compared with baselines. Imagine a test point (a fox image) has unreliable representations. It sits close to a reference point A (a dog image) and far from another point B (a cat image) in one representation space, while the opposite holds true in another representation space. In this case, $NC_k$ would assign a low score due to this inconsistency. However, $Dist_k$, LL, and Norm suggest "reliable" since they rely on the relative distance of the test point to the closest reference point (or to the origin). FV may also indicate "reliable" if the representations of the test point remain consistent despite falling into clusters of dog or cat images in different representation spaces. None of these baselines align different representation spaces before computing these relative distances.

### 4.2.2 Correlation to Transfer Learning Performance

We conduct experiments with out-of-distribution tasks, particularly focusing on transfer learning tasks. We use the `TinyImagenet` dataset [Le and Yang, 2015] as the source dataset for pre-training the embedding functions. Subsequently, we fine-tune and evaluate the embedding functions on three target datasets: `CIFAR-10`, `CIFAR-100`, and `STL-10` [Coates et al., 2011]. The correlation between each method and the representation reliability is reported in Table 2. As observed, NC consistently captures the representation reliability across a diverse range of settings.

We observe that Norm achieves comparable performance with NC. One possible explanation is that several factors (e.g., properties of embedding functions and characteristics

of test points) influence the reliability of a test point's representation. In the context of transfer learning, OOD test points are more likely to be unreliable. The baseline we selected, Norm, is good at identifying OOD samples [Tack et al., 2020], thereby enabling it to capture representation reliability to some extent.

## 4.3 USE CASE: RANK PRE-TRAINED MODELS

In practice, there are typically several off-the-shelf pre-training models available for downstream tasks. These models may vary in architecture and are trained using different learning paradigms, making it challenging to decide which one to use. We apply our $NC_{100}$ to aid in this selection process by ranking these models based on their average reliability scores. We extend the previous experiments on transfer learning scenarios as follows. For each data point, we rank the three pre-training models (SimCLR, BYOL, and MoCo) using $NC_{100}$ and baselines, respectively. Then we compute the correlation between these rankings and the actual ranking (based on average downstream tasks performance) by computing Kendall's $\tau$ coefficient score between the two rankings. We present the experimental results in Table 3.

Our NC demonstrates the second-best performance for ResNet-18 and the best performance for ResNet-50. In contrast, the baselines ($Dist_1$, Norm, and LL) exhibit negative scores when ranking embedding functions. The only exception is FV which achieves a comparable performance with NC, while demonstrating a low (or even negative) correlation in the previous experiments (Tables 1 and 2). One possible interpretation is that the primary issue with FV lies in its failure to align different representation spaces before comparing them. When ranking different embedding functions, the degrees of misalignment between their ensembles should roughly be of the same order, as these embedding functions share the same architecture. Consequently, this error term would cancel out when ranking these embedding functions.

## 4.4 QUANTIFYING THE RELIABILITY OF INDIVIDUAL EMBEDDING FUNCTIONS

Our NC requires a set $\mathcal{H}$ of embedding functions, comparing the consistent neighbors of a test point across the representation spaces generated by these functions. When $\mathcal{H}$ has a single function $h$, we can still use our NC by applying the same learning algorithm (with varying random seeds) and data used to train $h$, yielding additional embedding functions to augment $\mathcal{H}$. The rationale behind this is that these embedding functions will possess similar inductive biases and generate representation spaces with comparable reliability. We validate this intuition in Table 8 and Table 9 in Appendix C.2. The results demonstrate that we can effectively

Table 2: Comparison of our neighborhood consistency ($NC_{100}$) with baseline methods in terms of their correlation with the representation reliability. Here the downstream tasks are transfer learning tasks, where embedding functions are pre-trained on `TinyImagenet` and fine-tuned on `CIFAR-10`, `CIFAR-100`, and `STL-10`.

| Pretraining Algorithms | Method | ResNet-18 | | | | | | ResNet-50 | | | | | |
|---|---|---|---|---|---|---|---|---|---|---|---|---|---|
| | | `TinyImagenet` → | | | | | | `TinyImagenet` → | | | | | |
| | | CIFAR-10 | | CIFAR-100 | | STL-10 | | CIFAR-10 | | CIFAR-100 | | STL-10 | |
| | | Entropy | Brier | Entropy | Brier | Entropy | Brier | Entropy | Brier | Entropy | Brier | Entropy | Brier |
| SimCLR | $NC_{100}$ | 0.1729 | 0.1292 | **0.1680** | **0.1343** | **0.2176** | **0.1513** | 0.1224 | 0.0804 | **0.1467** | **0.1194** | **0.1866** | **0.1169** |
| | $Dist_1$ | 0.1311 | 0.0933 | 0.0138 | -0.0251 | 0.1098 | 0.0520 | 0.0894 | 0.0531 | -0.0246 | -0.0627 | 0.0340 | -0.0167 |
| | Norm | **0.2124** | **0.1642** | 0.1220 | 0.0507 | 0.1641 | 0.0852 | **0.1549** | **0.1098** | 0.0895 | 0.0177 | 0.0608 | -0.0056 |
| | LL | 0.0369 | 0.0168 | -0.0991 | -0.1269 | 0.0205 | -0.0180 | 0.0225 | 0.0027 | -0.1192 | -0.1443 | -0.0149 | -0.0495 |
| | FV | -0.0919 | -0.0779 | -0.1660 | -0.1668 | -0.1872 | -0.1593 | -0.1078 | -0.0878 | -0.1867 | -0.1840 | -0.2178 | -0.1829 |
| BYOL | $NC_{100}$ | **0.2557** | **0.1431** | **0.2352** | **0.1525** | 0.0374 | 0.0232 | **0.2561** | **0.1258** | **0.2600** | **0.1005** | **0.1410** | 0.0123 |
| | $Dist_1$ | -0.1458 | -0.0923 | -0.2716 | -0.1525 | -0.2893 | -0.2178 | -0.1885 | -0.1274 | -0.3075 | -0.1679 | -0.2371 | -0.1925 |
| | Norm | 0.1483 | 0.0749 | 0.1879 | 0.1315 | **0.1021** | **0.0772** | 0.1620 | 0.0918 | 0.1082 | 0.0816 | 0.0963 | **0.0380** |
| | LL | -0.2106 | -0.1248 | -0.3418 | -0.2006 | -0.3319 | -0.2472 | -0.3130 | -0.1763 | -0.3671 | -0.1797 | -0.3235 | -0.2002 |
| | FV | -0.1386 | -0.0713 | -0.1874 | -0.1202 | -0.1171 | -0.0790 | -0.1668 | -0.0737 | -0.1441 | -0.0705 | -0.0584 | 0.0205 |
| MoCo | $NC_{100}$ | 0.1880 | 0.1313 | **0.1797** | **0.1446** | 0.2263 | 0.1463 | 0.1927 | 0.1300 | **0.1962** | **0.1560** | 0.2170 | 0.1433 |
| | $Dist_1$ | 0.1213 | 0.0718 | 0.0311 | -0.0143 | 0.0993 | 0.0223 | 0.1751 | 0.1109 | 0.0759 | 0.0257 | 0.1696 | 0.0710 |
| | Norm | **0.1930** | **0.1355** | 0.1321 | 0.0531 | 0.1327 | 0.0443 | **0.2503** | **0.1803** | 0.1838 | 0.1034 | **0.2524** | **0.1524** |
| | LL | 0.0158 | -0.0082 | -0.1046 | -0.1386 | -0.0083 | -0.0577 | 0.0458 | 0.0145 | -0.0916 | -0.1239 | 0.0133 | -0.0502 |
| | FV | -0.0793 | -0.0701 | -0.1499 | -0.1658 | -0.1615 | -0.1577 | -0.0560 | -0.0400 | -0.1510 | -0.1578 | -0.1735 | -0.1577 |

Table 3: Comparison of our $NC_{100}$ with baselines in selecting reliable backbones among those pre-trained by three algorithms (SimCLR, BYOL, and MOCO) on the `TinyImagenet` dataset. For each test point within the `CIFAR-10`, `CIFAR-100`, and `STL-10` datasets, we calculate Kendall's $\tau$ coefficient to evaluate the correlation between the rankings given by each method and the rankings given by the actual downstream performance.

| Method | ResNet-18 | | | | | | ResNet-50 | | | | | |
|---|---|---|---|---|---|---|---|---|---|---|---|---|
| | `TinyImagenet` → | | | | | | `TinyImagenet` → | | | | | |
| | CIFAR-10 | | CIFAR-100 | | STL-10 | | CIFAR-10 | | CIFAR-100 | | STL-10 | |
| | Entropy | Brier | Entropy | Brier | Entropy | Brier | Entropy | Brier | Entropy | Brier | Entropy | Brier |
| $NC_{100}$ | 0.4841 | 0.4625 | 0.4211 | 0.4199 | 0.2342 | 0.3073 | **0.5051** | **0.5089** | **0.4865** | **0.4912** | **0.4512** | **0.4385** |
| $Dist_1$ | -0.3328 | -0.3621 | -0.2863 | -0.3364 | -0.0457 | -0.1729 | -0.3552 | -0.4181 | -0.3196 | -0.3869 | -0.2162 | -0.2719 |
| Norm | -0.3199 | -0.3543 | -0.2619 | -0.3198 | -0.0342 | -0.1654 | -0.3543 | -0.4175 | -0.3172 | -0.3851 | -0.2154 | -0.2712 |
| LL | -0.3259 | -0.3575 | -0.2776 | -0.3296 | -0.0430 | -0.1692 | -0.3615 | -0.4220 | -0.3333 | -0.3963 | -0.2228 | -0.2764 |
| FV | **0.8988** | **0.7348** | **0.8321** | **0.6624** | **0.7027** | **0.6186** | 0.3568 | 0.2891 | 0.2207 | 0.1468 | 0.2738 | 0.2130 |

predict the reliability of individual embedding functions using this approach.

reliability. In contrast, $NC_{100}$ consistently demonstrates a positive correlation and ranks within the top 2 among all baseline methods, regardless of the distance metric chosen.

## 4.5 ABLATION STUDIES

**Robustness to the Choice of Distance Metric.** We explore how various distance metrics within the representation space affect both our method and the baseline approaches. While Euclidean distance is a natural choice for the representations in $\mathbb{R}^d$, we also investigate cosine distance. This choice is motivated by the widespread use of cosine similarity in self-supervised algorithms, such as SimCLR, within their loss functions.

We present the results in Table 5 and 7 in Appendix C.1. Our key observation is: the baselines, including $Dist_1$ and LL, are sensitive to the choice of distance metric and may even exhibit negative correlations with the representation

**Performance with a Smaller Ensemble Size ($M$).** We conduct an ablation study to investigate the impact of ensemble size $M$ on estimating the representation reliability. The results in Figure 2(a) and 2(b) and Appendix C show that increasing ensemble size improves the correlation scores. Nonetheless, our method generally outperforms baseline approaches, even with a small ensemble size of $M = 2$. Exploring cost-effective ensemble construction methods, such as low-rank approximations [Wen et al., 2020] or stochastic weight averaging [Izmailov et al., 2018, Maddox et al., 2019], can be a promising future direction.

**Trade-off on the Number of Neighbors ($k$).** As discussed in Section 3.3, the choice of $k$ in Equation (7) leads

to a trade-off between having more consistent neighbors and preserving the overall reliability of those neighbors. In order to explore this trade-off, we conduct experiments with different values of $k \in \{1, 2, 5, 10, 20, 50, 100, 200, 500, 1000\}$. The correlation between our method and the representation reliability is illustrated in Figure 2(c) and 2(d) and Appendix C: it initially increases and then decreases as expected. We observe that the optimal performance could be achieved with $k$ around 100 (i.e., 2% of $|\boldsymbol{X}_{\text{ref}}| = 5000$) for $\mathsf{NC}_k$ across different pre-training algorithms, model architectures, and downstream data.

## 4.6 KEY OBSERVATIONS & TAKEAWAYS

In summary, our main findings from the experiments are:

- Our proposed method $\mathsf{NC}_k$ effectively captures the representation reliability and can help compare the reliability of different pre-trained models.

- More importantly, contrary to the baselines, $\mathsf{NC}_k$ *consistently* exhibits a positive correlation with the representation reliability across all different settings. Although the baselines may occasionally surpass $\mathsf{NC}_k$, their performance fluctuates significantly across different settings and sometimes becomes even negative, introducing a risk when used to assess reliability in safety-critical settings.

## 5 FINAL REMARKS AND LIMITATIONS

Self-supervised learning is increasingly used for training general-purpose embedding functions that can be adapted to various downstream tasks. In this paper, we presented a systematic study to evaluate the quality of representations assigned by the embedding functions. We introduced a mathematical definition of representation reliability, demonstrated that existing UQ frameworks in supervised learning cannot be directly applied to estimate uncertainty in representation spaces, derived an estimate for the representation reliability, and validated our estimate through extensive numerical experiments.

There is a crucial need for future research to investigate and ensure the responsibility and trustworthiness of pre-trained self-supervised models. For example, representations should be interpretable and not compromise private information. Moreover, the embedding functions should exhibit robustness against adversarial attacks and incorporate a notion of uncertainty, in addition to their abstract representations. This work takes an initial step towards understanding the uncertainty of the representations. In the case where a downstream model fails to deliver a desirable output for a test point, the representation reliability can provide valuable insight into whether the mistake is due to unreliable representations or downstream heads.

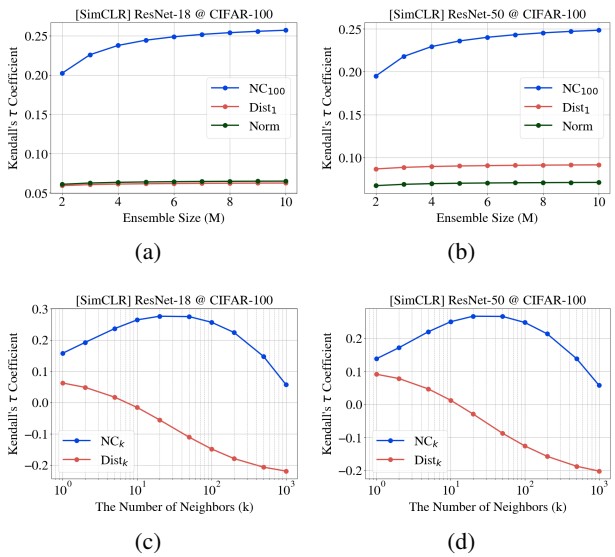

Figure 2: Ablation studies on the ensemble size ($M$) and the number of neighbors ($k$) for $\mathsf{NC}_k$ (ours) and baselines. Brier score is used for the downstream performance metric. The comprehensive results can be found in Appendix C.1.

There are several future directions that are worth further exploration. For example, our current method for estimating the representation reliability uses a set of embedding functions to compute neighborhood consistency. It would be interesting to investigate whether our approach can be expanded to avoid the need for training multiple embedding functions. This could potentially be achieved through techniques such as MC dropout or adding random noise to the neural network parameters in order to perturb them slightly. Additionally, while we currently assess the representation reliability through downstream prediction tasks, it would be valuable to investigate the extension of our definition to cover a broader range of downstream tasks.

### Acknowledgements

The authors would like to thank Dr. Akash Srivastava for insightful discussions. This work was supported in part by the MIT-IBM Watson AI Lab, MathWorks, and Amazon. The authors also acknowledge the MIT SuperCloud and Lincoln Laboratory Supercomputing Center for providing computing resources that have contributed to the results reported within this paper.

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

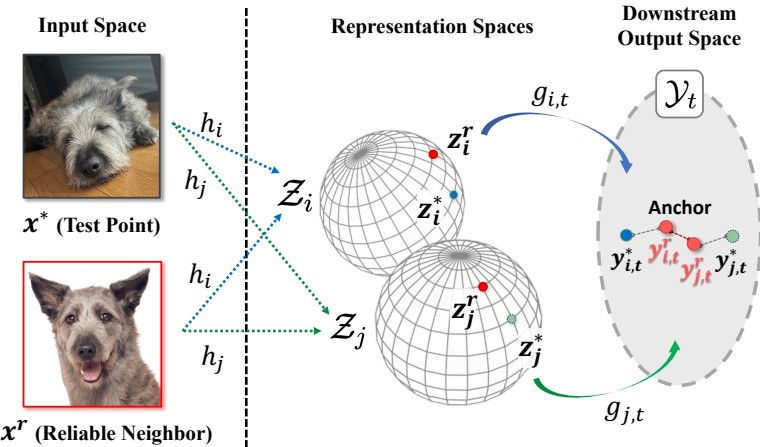

Figure 3: Graphical visualization of the sketch for the proof of Theorem 2. Let $\mathcal{Z}i$ and $\mathcal{Z}j$ denote the representation spaces defined by the embedding functions $h_i$ and $h_j$, respectively. Suppose that there is a reliable neighboring point $\boldsymbol{x}^r$ that is located close to the test point $\boldsymbol{x}^*$ in each representation space. For any downstream task $t$, a reliable neighboring point $\boldsymbol{x}^r$ serves as an anchor for comparing different representations $\boldsymbol{z}_i^* = h_i(\boldsymbol{x}^*)$ and $\boldsymbol{z}_j^* = h_j(\boldsymbol{x}^*)$ of the test point $\boldsymbol{x}^*$. The key idea is that $y_{i,t}^*$ and $y_{j,t}^*$ — the downstream predictions on the test point using the two different embedding functions — should be similar because the predictions $y_{i,t}^*$ and $y_{i,t}^r$ as well as $y_{j,t}^r$ and $y_{j,t}^*$ are similar due to the Lipschitz continuity of the downstream predictors. Additionally, since $\boldsymbol{x}^r$ is a reliable point, the predictions $y_{i,t}^r$ and $y_{j,t}^r$ are similar. Thus, it follows that $y_{i,t}^*$ and $y_{j,t}^*$ are similar as well.

# A    OMITTED PROOFS

## A.1    PROOF OF THEOREM 1

We present a more rigorous statement of Theorem 1 along with its proof.

**Theorem 3.** *Consider downstream tasks as prediction tasks, with the class of downstream heads closed under linear transformations (i.e., if $g$ belongs to this class, $g \circ \boldsymbol{A}$ also belongs to it for any matrix $\boldsymbol{A}$). Let* $\mathsf{diam}(\mathcal{Z}) = \sup_{\boldsymbol{z},\boldsymbol{z}' \in \mathcal{Z}} \|\boldsymbol{z} - \boldsymbol{z}'\|_2$ *be the diameter of the representation space. Then for any $A < (\mathsf{diam}(\mathcal{Z})/2)^2$ and a test point $\boldsymbol{x}^*$, there exist embedding functions $h_1, \cdots, h_{2M} \in \mathcal{H}$ such that* $\mathsf{Var}_{i \sim [2M]}\left(h_i(\boldsymbol{x}^*)\right) \geq A$ *but* $\mathsf{Var}_{i \sim [2M]}\left(g_{i,t} \circ h_i(\boldsymbol{x}^*)\right) = 0$ *for any downstream task $t$, where $g_{i,t}$ is an optimal downstream head for $h_i$ under task $t$.*

*Proof.* We prove this theorem via construction, assuming that there are only two embedding functions $h_1$, $h_2$, without loss of generality. Let $\delta > 0$ be a small constant. We select two points $\boldsymbol{z}_1, \boldsymbol{z}_2 \in \mathcal{Z}$ such that $\|\boldsymbol{z}_1 - \boldsymbol{z}_2\|_2 \geq \mathsf{diam}(\mathcal{Z}) - \delta$. Then we define $h_1$ and $h_2$ to satisfy the condition: $h_i(\boldsymbol{x}^*) = \boldsymbol{z}_i$ and $h_1 = \boldsymbol{A} \circ h_2$ where $\boldsymbol{A}$ is an invertible matrix. Consequently,

$$\mathsf{Var}_{i \sim [2]}\left(h_i(\boldsymbol{x}^*)\right) = \left\|\frac{\boldsymbol{z}_1 - \boldsymbol{z}_2}{2}\right\|_2^2 \geq \left(\frac{\mathsf{diam}(\mathcal{Z}) - \delta}{2}\right)^2.$$

For a given task $t$, let $g_{1,t}$ be an optimal downstream head for the embedding function $h_1$. Next, we prove that $g_{1,t} \circ \boldsymbol{A}$ is an optimal downstream head for $h_2$ under the same task. Suppose otherwise, if there exists another downstream head $g'_{2,t}$ achieving higher performance than $g_{1,t} \circ \boldsymbol{A}$ when combined with $h_2$. Since $h_1 = \boldsymbol{A} \circ h_2$, it implies $g'_{2,t} \circ \boldsymbol{A}^{-1}$ can achieve higher performance than $g_{1,t}$ when combined with $h_1$. This contradicts the optimally of $g_{1,t}$. Therefore, $g_{1,t} \circ \boldsymbol{A}$ is an optimal downstream head for $h_2$ for task $t$ so we denote it as $g_{2,t}$. Now, we have

$$g_{2,t} \circ h_2(\boldsymbol{x}^*) = g_{1,t} \circ \boldsymbol{A} \circ h_2(\boldsymbol{x}^*) = g_{1,t} \circ \boldsymbol{A} \circ \boldsymbol{A}^{-1} \circ h_1(\boldsymbol{x}^*) = g_{1,t} \circ h_1(\boldsymbol{x}^*),$$

which leads to $\mathsf{Var}_{i \sim [2]}\left(g_{i,t} \circ h_i(\boldsymbol{x}^*)\right) = 0$. As $\delta$ can be chosen arbitrarily small, we can achieve the desired result. By setting $h_{2i-1}(\boldsymbol{x}^*) = \boldsymbol{z}_1$ and $h_{2i}(\boldsymbol{x}^*) = \boldsymbol{z}_2$, the same proof can be extended to the case of $2M$ case. □

## A.2 PROOF OF THEOREM 2

*Proof.* For the sake of notational simplicity, we denote $f_i = g_{i,t} \circ h_i$. When $g_{i,t}$ is Lipschitz continuous (see Appendix A.3 for more detail) and Equation (5) — $\|h_i(\boldsymbol{x}^r) - h_i(\boldsymbol{x}^*))\|_2 \leq \epsilon_{nb}$ — holds, we have:

$$\|f_i(\boldsymbol{x}^r) - f_i(\boldsymbol{x}^*)\|_2 \leq L_{i,t} \cdot \|h_i(\boldsymbol{x}^r) - h_i(\boldsymbol{x}^*)\|_2 \leq L_{i,t} \cdot \epsilon_{nb} \leq L_t \cdot \epsilon_{nb} \ , \ \forall i \in [M]. \tag{8}$$

By the triangle inequality, we have the following upper bound for the output difference:

$$\begin{aligned}
\|f_i(\boldsymbol{x}^*) - f_j(\boldsymbol{x}^*)\|_2^2 &\leq \left(\|f_i(\boldsymbol{x}^*) - f_i(\boldsymbol{x}^r)\|_2 + \|f_i(\boldsymbol{x}^r) - f_j(\boldsymbol{x}^r)\|_2 + \|f_j(\boldsymbol{x}^r) - f_j(\boldsymbol{x}^*)\|_2\right)^2 \\
&\leq \left(L_t\epsilon_{nb} + \|f_i(\boldsymbol{x}^r) - f_j(\boldsymbol{x}^r)\|_2 + L_t\epsilon_{nb}\right)^2 \\
&= 4\left(L_t\epsilon_{nb}\right)^2 + 4\left(L_t\epsilon_{nb}\right)\underbrace{\|f_i(\boldsymbol{x}^r) - f_j(\boldsymbol{x}^r)\|_2}_{L_2 \text{ difference}} + \underbrace{\|f_i(\boldsymbol{x}^r) - f_j(\boldsymbol{x}^r)\|_2^2}_{L_2\text{-squared difference}}.
\end{aligned} \tag{9}$$

Note that the ensemble variance is proportional to the average pairwise $L_2$-squared difference across the ensemble of $f_i$:

$$\text{Var}_{i\sim[M]}\left(g_{i,t} \circ h_i(\boldsymbol{x}^*)\right) = \frac{1}{M^2}\sum_{i<j}\|f_i(\boldsymbol{x}^*) - f_j(\boldsymbol{x}^*)\|_2^2 \tag{10}$$

$$\text{Var}_{i\sim[M]}\left(g_{i,t} \circ h_i(\boldsymbol{x}^r)\right) = \frac{1}{M^2}\sum_{i<j}\|f_i(\boldsymbol{x}^r) - f_j(\boldsymbol{x}^r)\|_2^2 \equiv \sigma_{r,t}^2. \tag{11}$$

Furthermore, by Cauchy-Schwarz inequality, the average pairwise $L_2$ difference is bounded by:

$$\begin{aligned}
\frac{1}{M^2}\sum_{i<j}\|f_i(\boldsymbol{x}^r) - f_j(\boldsymbol{x}^r)\|_2 &\leq \frac{1}{M^2}\left(\sum_{i<j}\|f_i(\boldsymbol{x}^r) - f_j(\boldsymbol{x}^r)\|_2^2\right)^{1/2} \cdot \left(\sum_{i<j}1\right)^{1/2} \\
&\leq \frac{1}{M^2}\left(M\sigma_{r,t}\right) \cdot \left(\frac{M(M-1)}{2}\right)^{1/2} \leq \frac{\sqrt{2}}{2}\sigma_{r,t}
\end{aligned} \tag{12}$$

Thus:

$$\begin{aligned}
\text{Var}_{i\sim[M]}\left(g_{i,t} \circ h_i(\boldsymbol{x}^*)\right) &= \frac{1}{M^2}\sum_{i<j}\|f_i(\boldsymbol{x}^*) - f_j(\boldsymbol{x}^*)\|_2^2 \\
&\leq \left(\frac{1}{M^2}\right)\left(\frac{M(M-1)}{2}\right)\left(4\left(L_t\epsilon_{nb}\right)^2\right) + 2\sqrt{2}\left(L_t\epsilon_{nb}\right)\sigma_{r,t} + \sigma_{r,t}^2 \\
&\leq \left(\sqrt{2}L_t\epsilon_{nb}\right)^2 + 2\sqrt{2}L_t\epsilon_{nb}\sigma_{r,t} + \sigma_{r,t}^2 \\
&= \left(\sqrt{2}L_t\epsilon_{nb} + \sigma_{r,t}\right)^2.
\end{aligned} \tag{13}$$

$\square$

The reader is referred to Figure 3 for a visualization accompanying the proof sketch.

## A.3 LIPSCHITZ CONTINUITY OF NEURAL NETWORKS

**Lemma 1.** *For a fully-connected layer $g_l(\boldsymbol{z}) = a(\boldsymbol{w}_l^T\boldsymbol{z} + \boldsymbol{b}_l)$ of downstream predictor, the similarity of outputs is bounded above by the Lipschitz continuity:*

$$\|g_l(\boldsymbol{z}) - g_l(\boldsymbol{z}^*)\|_2 \leq \|\boldsymbol{w}_l\|_2 \cdot \|\boldsymbol{z} - \boldsymbol{z}^*\|_2. \tag{14}$$

*Proof.* Suppose $a(\cdot)$ is 1-Lipschitz continuous (e.g., identity, ReLU, sigmoid, and softmax) ,

$$\begin{aligned}
\|g_l(\boldsymbol{z}) - g_l(\boldsymbol{z}^*)\|_2 &= \|a\left(\boldsymbol{w}_l^T\boldsymbol{z} + \boldsymbol{b}_l\right) - a\left(\boldsymbol{w}_l^T\boldsymbol{z}^* + \boldsymbol{b}_l\right)\|_2 \\
&\leq 1 \cdot \|\left(\boldsymbol{w}_l^T\boldsymbol{z} + \boldsymbol{b}_l\right) - \left(\boldsymbol{w}_l^T\boldsymbol{z}^* + \boldsymbol{b}_l\right)\|_2 \\
&= \|\boldsymbol{w}_l^T\left(\boldsymbol{z} - \boldsymbol{z}^*\right)\|_2 \\
&\leq \|\boldsymbol{w}_l\|_2 \cdot \|\boldsymbol{z} - \boldsymbol{z}^*\|_2.
\end{aligned} \tag{15}$$

where $\|\cdot\|_2$ is the spectral norm for a matrix and $L_2$ norm for a vector. $\square$

**Corollary 1.** *For a feed-forward neural networks, $g = g_1 \circ g_2 \circ ... \circ g_L$, composed of multiple fully-connected layers, the similarity of predictions at $\boldsymbol{x}$ and $\boldsymbol{x}^*$ is bounded above by:*

$$\|g(\boldsymbol{x}) - g(\boldsymbol{x}^*)\|_2 \le L_t \cdot \|\boldsymbol{x} - \boldsymbol{x}^*\|_2. \tag{16}$$

*where $L_t = \|\boldsymbol{w}_1\|_2 \cdot \|\boldsymbol{w}_2\|_2 \cdots \|\boldsymbol{w}_L\|_2$ is a Lipschitz constant for g.*

## B   COMPUTING REPRESENTATION RELIABILITY

Recall that the representation reliability is defined by:

$$\mathsf{Reli}(\boldsymbol{x}^*; \mathcal{H}, \mathcal{T}) \triangleq \sum\nolimits_{t \in \mathcal{T}} \mathsf{Perf}_t\left(g_{h,t} \circ h,\ \boldsymbol{x}^*\right) /\ |\mathcal{T}|$$

where $\mathcal{T}$ is the collection of downstream tasks and for each task $t \in \mathcal{T}$, an embedding function $h$ is taken uniformly at random from $\mathcal{H}$ and a downstream predictor $g_{h,t}$ is (optimally) trained upon $h$ on $t$. In this section, we provide some examples to compute representation reliability using standard uncertainty or accuracy measures using an ensemble of embedding functions $\{h_1, \cdots, h_M\}$ and the corresponding downstream predictors $\{g_{1,t}, \cdots, g_{M,t}\}$.

Let us denote this distribution of $h$: $\mathcal{P}_{\mathcal{H}}$. Consequently, in regression tasks, the negative variance of the predictive distribution can be utilized for a performance metric:

$$
\begin{aligned}
\mathsf{Perf}_t\left(g_{h,t} \circ h,\ \boldsymbol{x}^*\right) &:= -\mathsf{Var}\left(g_{h,t} \circ h(\boldsymbol{x}^*)\right) \\
&= -\mathbb{E}_{h \sim \mathcal{P}_{\mathcal{H}}}\left[\left(g_{h,t} \circ h(\boldsymbol{x}^*) - \mathbb{E}_{h \sim \mathcal{P}_{\mathcal{H}}}\left[g_{h,t} \circ h(\boldsymbol{x}^*)\right]\right)^2\right] \\
&\approx -\frac{1}{M^2} \sum_{i<j} \left(g_{i,t} \circ h_i(\mathbf{x}^*) - g_{j,t} \circ h_j(\mathbf{x}^*)\right)^2
\end{aligned}
\tag{17}
$$

where $g_{i,t} \equiv g_{h_i,t}$. In the context of multi-output tasks, the trace of the covariance matrix can be utilized, which is essentially the sum of variances across each output dimension. This approach aggregates the individual uncertainties of each output, providing a comprehensive measure of overall uncertainty in the multi-output setting.

While our theoretical analysis has focused on regression tasks using variance, for classification tasks, metrics like the Brier score or entropy are more natural choices. For a given downstream classification task $t$ with $C$ classes, the ground truth label of $\boldsymbol{x}^*$ is represented by a one-hot vector: $y_t^* = (y_{t[1]}^*, \cdots, y_{t[C]}^*)^T$ where $y_{t[c]}^* \in \{0, 1\}$. The softmax output for class $c$ via the predictive function $g_i \circ h_i(\cdot)$ is denoted as $g_i \circ h_i(\cdot)_{[c]}$.

The negative Brier score for this setup is calculated as follows:

$$
\begin{aligned}
\mathsf{Perf}_t\left(g_{h,t} \circ h,\ \boldsymbol{x}^*\right) &:= -\mathsf{Brier}(\mathbb{E}_{h \sim \mathcal{P}_{\mathcal{H}}}\left[g_{h,t} \circ h(\boldsymbol{x}^*)\right]; y_t^*) \\
&= -\sum_{c=1}^{C} \left(\mathbb{E}_{h \sim \mathcal{P}_{\mathcal{H}}}\left[g_{h,t} \circ h(\mathbf{x}^*)\right]_{[c]} - y_{t[c]}^*\right)^2
\end{aligned}
\tag{18}
$$

$$
-\approx \sum_{c=1}^{C} \left(\frac{1}{M} \sum_{i=1}^{M} g_{i,t} \circ h_i(\mathbf{x}^*)_{[c]} - y_{t[c]}^*\right)^2
\tag{19}
$$

and the negative entropy is given by:

$$
\begin{aligned}
\mathsf{Perf}_t\left(g_{h,t} \circ h,\ \boldsymbol{x}^*\right) &:= -\mathsf{Entropy}(\mathbb{E}_{h \sim \mathcal{P}_{\mathcal{H}}}\left[g_{h,t} \circ h(\mathbf{x}^*)\right]) \\
&\approx \sum_{c=1}^{C} \left(\frac{1}{M} \sum_{i=1}^{M} g_{i,t} \circ h_i(\mathbf{x}^*)_{[c]}\right) \log\left(\frac{1}{M} \sum_{i=1}^{M} g_{i,t} \circ h_i(\mathbf{x}^*)_{[c]}\right).
\end{aligned}
\tag{20}
$$

While establishing a rigorous relationship between uncertainty and predictive accuracy may pose challenges, empirical studies have demonstrated a strong correlation between the two measures. As stated in Theorem 2, a lower bound of the representation reliability measured by $\mathsf{Perf}_t(\cdot) := -\mathsf{Var}(\cdot)$, depicted in Equation (17), is assured by the sum of the reference point's reliability along with its relative distance to the test point. Consequently, this explains how the NC score could effectively capture representation reliability, irrespective of the specific performance metrics employed.

# C ADDITIONAL EXPERIMENTAL RESULTS

In the following Appendices, we include the standard deviation resulting from the randomness involved in selecting the reference dataset, as well as the averaged values.

## C.1 ABLATION STUDIES

**Unnormalized representation.** In the main manuscript, we normalize the representation provided by the embedding function before computing our NC and baselines: $h(\boldsymbol{x})/\|h(\boldsymbol{x})\|_2 \in \mathcal{S}^{d-1}$. Here we evaluate the performance of these methods using the original, unnormalized representations. For our $\mathsf{NC}_k$ and $\mathsf{Dist}_k$, we use Euclidean distance as a distance metric in the representation space. For Norm, we directly compute the $L_2$ norm of the representations. For FV, we calculate the variance of the representations in this unnormalized representation space. For LL, we fit a Gaussian mixture model and compute the log-likelihood of this model. We reproduce the results from the main manuscript's Table 1 (ID) and Table 2 (Transfer). The results are shown in Table 5 and Table 7, respectively.

As shown in Table 5 (ID), Table 7 (Transfer), our method exhibits consistent and strong performance, even when applying directly to the unnormalized representation space. On the other hand, the baseline methods suffer from a significant performance reduction when paired with Euclidean distance. Our conjecture is as follows. As noted in our results and Tack et al. [2020], points with a larger $L_2$-norm tend to exhibit higher reliability. This implies that reliable points are often located in the outer regions of the representation space, resulting in larger distances between points in those regions compared to points near the origin. This observation, however, contradicts to the $\mathsf{Dist}_1$ and FV's expectations (i.e., smaller value is presumed to indicate higher reliability). As a result, when these measures are applied to the unnormalized representations, they struggle to accurately reflect the representation reliability, potentially indicating negative correlations that defy their assumptions.

**Ensemble sizes ($M$) and the number of neighbors ($k$).** In the main manuscript, we present an ablation study to investigate the impact of ensemble size $M$ and the number of neighbors $k$ on estimating the representation reliability (see Figure 2). Here we provide additional ablation results in Figure 4 (for the impact of $M$) and Figure 5 (for the impact of $k$). These results cover various combinations of pre-training algorithms, model architectures, and datasets. The Brier score is used in all experimental results to measure downstream task performance.

## C.2 QUANTIFYING THE RELIABILITY OF INDIVIDUAL EMBEDDING FUNCTIONS

To confirm whether the NC score can also be applied to gauge the reliability of the individual embedding function $h \in \mathcal{H}$, we measured the correlation between the measures and the representation reliability estimated with Brier score (i.e., downstream accuracy), for each member of the ensemble. The results, detailed in Table 8 and Table 9, include the average correlation and its standard deviation across the individuals. Since metrics such as $\mathsf{Dist}_1$, norm, and LL are inherently applicable to individual embedding functions without using the ensemble, we also report non-ensemble scores as well, marked with a superscript star (*).

Table 4: Comparison on the correlation between our neighborhood consistency ($NC_{100}$) and baseline methods in relation to performance on in-distribution downstream tasks.

| Pretraining Algorithms | Method | ResNet-18 | | | |
| --- | --- | --- | --- | --- | --- |
| | | CIFAR-10 | | CIFAR-100 | |
| | | Entropy | Brier | Entropy | Brier |
| SimCLR | $NC_{100}$ | **0.3865** $_{\pm 0.0024}$ | **0.3262** $_{\pm 0.0022}$ | **0.3130** $_{\pm 0.0031}$ | **0.2590** $_{\pm 0.0028}$ |
| | $Dist_1$ | 0.3021 $_{\pm 0.0063}$ | 0.2544 $_{\pm 0.0051}$ | 0.0757 $_{\pm 0.0084}$ | 0.0635 $_{\pm 0.0075}$ |
| | Norm | 0.2907 | 0.2429 | 0.1198 | 0.0649 |
| | LL | 0.1797 $_{\pm 0.0078}$ | 0.1356 $_{\pm 0.0068}$ | -0.0989 $_{\pm 0.0063}$ | -0.1067 $_{\pm 0.0056}$ |
| | FV | -0.0476 | -0.0458 | -0.1891 | -0.1710 |
| BYOL | $NC_{100}$ | **0.2749** $_{\pm 0.0049}$ | **0.1826** $_{\pm 0.0030}$ | **0.2354** $_{\pm 0.0048}$ | **0.1753** $_{\pm 0.0042}$ |
| | $Dist_1$ | 0.0322 $_{\pm 0.0029}$ | 0.0385 $_{\pm 0.0018}$ | -0.1477 $_{\pm 0.0045}$ | -0.0709 $_{\pm 0.0031}$ |
| | Norm | 0.0725 | 0.0235 | 0.1659 | 0.1231 |
| | LL | -0.0539 $_{\pm 0.0010}$ | -0.0196 $_{\pm 0.0006}$ | -0.2637 $_{\pm 0.0052}$ | -0.1650 $_{\pm 0.0045}$ |
| | FV | 0.1439 | 0.1003 | -0.1727 | -0.1284 |
| MoCo | $NC_{100}$ | **0.4157** $_{\pm 0.0042}$ | **0.3653** $_{\pm 0.0038}$ | **0.3632** $_{\pm 0.0019}$ | **0.3128** $_{\pm 0.0024}$ |
| | $Dist_1$ | 0.3523 $_{\pm 0.0048}$ | 0.3093 $_{\pm 0.0037}$ | 0.2246 $_{\pm 0.0056}$ | 0.1925 $_{\pm 0.0070}$ |
| | Norm | 0.3238 | 0.2708 | 0.2186 | 0.1432 |
| | LL | 0.2201 $_{\pm 0.0108}$ | 0.1744 $_{\pm 0.0111}$ | -0.0334 $_{\pm 0.0052}$ | -0.0736 $_{\pm 0.0038}$ |
| | FV | 0.0519 | 0.0321 | -0.1420 | -0.1578 |

| Pretraining Algorithms | Method | ResNet-50 | | | |
| --- | --- | --- | --- | --- | --- |
| | | CIFAR-10 | | CIFAR-100 | |
| | | Entropy | Brier | Entropy | Brier |
| SimCLR | $NC_{100}$ | **0.3786** $_{\pm 0.0013}$ | **0.3206** $_{\pm 0.0014}$ | **0.3070** $_{\pm 0.0040}$ | **0.2516** $_{\pm 0.0036}$ |
| | $Dist_1$ | 0.2750 $_{\pm 0.0052}$ | 0.2376 $_{\pm 0.0044}$ | 0.1100 $_{\pm 0.0060}$ | 0.0919 $_{\pm 0.0061}$ |
| | Norm | 0.2789 | 0.2385 | 0.1311 | 0.0710 |
| | LL | 0.1700 $_{\pm 0.0088}$ | 0.1348 $_{\pm 0.0073}$ | -0.0831 $_{\pm 0.0036}$ | -0.0959 $_{\pm 0.0032}$ |
| | FV | -0.0402 | -0.0393 | -0.1984 | -0.1870 |
| BYOL | $NC_{100}$ | **0.3524** $_{\pm 0.0028}$ | **0.2131** $_{\pm 0.0014}$ | **0.3233** $_{\pm 0.0061}$ | **0.1733** $_{\pm 0.0039}$ |
| | $Dist_1$ | -0.1858 $_{\pm 0.0022}$ | -0.1442 $_{\pm 0.0010}$ | -0.3124 $_{\pm 0.0019}$ | -0.1598 $_{\pm 0.0014}$ |
| | Norm | 0.1990 | 0.1184 | 0.1990 | 0.1476 |
| | LL | -0.2542 $_{\pm 0.0047}$ | -0.1784 $_{\pm 0.0035}$ | -0.3777 $_{\pm 0.0018}$ | -0.2018 $_{\pm 0.0013}$ |
| | FV | 0.0676 | 0.0524 | -0.0213 | -0.0625 |
| MoCo | $NC_{100}$ | 0.3757 $_{\pm 0.0041}$ | 0.3236 $_{\pm 0.0038}$ | **0.2667** $_{\pm 0.0043}$ | **0.2225** $_{\pm 0.0028}$ |
| | $Dist_1$ | **0.3831** $_{\pm 0.0032}$ | **0.3397** $_{\pm 0.0027}$ | 0.2453 $_{\pm 0.0051}$ | 0.2183 $_{\pm 0.0054}$ |
| | Norm | 0.3798 | 0.3294 | 0.2172 | 0.1659 |
| | LL | 0.2518 $_{\pm 0.0073}$ | 0.2125 $_{\pm 0.0073}$ | -0.0156 $_{\pm 0.0062}$ | -0.0343 $_{\pm 0.0054}$ |
| | FV | 0.1228 | 0.1122 | -0.1081 | -0.1100 |

Table 5: Comparison on the correlation between our neighborhood consistency ($NC_{100}$) and baseline methods in relation to performance on in-distribution downstream tasks, for the **unnormalized** representation. The overall correlation is weaker compared to the one observed when using normalized representation as presented in Table 4. Nevertheless, our approach consistently shows a robust performance compared to baseline methods.

| Pretraining Algorithms | Method | ResNet-18 | | | |
|---|---|---|---|---|---|
| | | CIFAR-10 | | CIFAR-100 | |
| | | Entropy | Brier | Entropy | Brier |
| SimCLR | $NC_{100}$ | **0.3237** $\pm$ 0.0023 | **0.2727** $\pm$ 0.0021 | **0.2868** $\pm$ 0.0014 | **0.2408** $\pm$ 0.0021 |
| | $Dist_1$ | -0.0590 $\pm$ 0.0071 | -0.0460 $\pm$ 0.0065 | -0.0317 $\pm$ 0.0061 | 0.0027 $\pm$ 0.0064 |
| | Norm | 0.2907 | 0.2429 | 0.1198 | 0.0649 |
| | LL | -0.0970 $\pm$ 0.0297 | -0.0830 $\pm$ 0.0276 | -0.2125 $\pm$ 0.0170 | -0.1704 $\pm$ 0.0142 |
| | FV | -0.3224 | -0.2704 | -0.2370 | -0.1723 |
| BYOL | $NC_{100}$ | **0.2381** $\pm$ 0.0051 | **0.1643** $\pm$ 0.0037 | **0.2284** $\pm$ 0.0044 | **0.1720** $\pm$ 0.0040 |
| | $Dist_1$ | -0.0801 $\pm$ 0.0016 | -0.0144 $\pm$ 0.0022 | -0.2077 $\pm$ 0.0038 | -0.1149 $\pm$ 0.0025 |
| | Norm | 0.0725 | 0.0235 | 0.1659 | 0.1231 |
| | LL | -0.1625 $\pm$ 0.0054 | -0.0695 $\pm$ 0.0049 | -0.3015 $\pm$ 0.0042 | -0.1955 $\pm$ 0.0038 |
| | FV | -0.0700 | -0.0228 | -0.1734 | -0.1286 |
| MoCo | $NC_{100}$ | **0.3739** $\pm$ 0.0068 | **0.3181** $\pm$ 0.0067 | **0.3273** $\pm$ 0.0042 | **0.2816** $\pm$ 0.0032 |
| | $Dist_1$ | -0.0741 $\pm$ 0.0049 | -0.0506 $\pm$ 0.0046 | -0.0305 $\pm$ 0.0060 | 0.0137 $\pm$ 0.0068 |
| | Norm | 0.3238 | 0.2708 | 0.2186 | 0.1432 |
| | LL | -0.1630 $\pm$ 0.0152 | -0.1421 $\pm$ 0.0145 | -0.2103 $\pm$ 0.0284 | -0.1786 $\pm$ 0.0246 |
| | FV | -0.3410 | -0.2899 | -0.3155 | -0.2432 |

| Pretraining Algorithms | Method | ResNet-50 | | | |
|---|---|---|---|---|---|
| | | CIFAR-10 | | CIFAR-100 | |
| | | Entropy | Brier | Entropy | Brier |
| SimCLR | $NC_{100}$ | **0.3366** $\pm$ 0.0062 | **0.2833** $\pm$ 0.0055 | **0.2635** $\pm$ 0.0040 | **0.2203** $\pm$ 0.0033 |
| | $Dist_1$ | -0.0445 $\pm$ 0.0056 | -0.0337 $\pm$ 0.0052 | -0.0328 $\pm$ 0.0056 | 0.0053 $\pm$ 0.0055 |
| | Norm | 0.2789 | 0.2385 | 0.1311 | 0.0710 |
| | LL | 0.0047 $\pm$ 0.0220 | 0.0053 $\pm$ 0.0205 | -0.1066 $\pm$ 0.0104 | -0.0739 $\pm$ 0.0097 |
| | FV | -0.3155 | -0.2706 | -0.2472 | -0.1833 |
| BYOL | $NC_{100}$ | **0.3704** $\pm$ 0.0021 | **0.2291** $\pm$ 0.0012 | **0.3194** $\pm$ 0.0061 | **0.1737** $\pm$ 0.0034 |
| | $Dist_1$ | -0.2229 $\pm$ 0.0023 | -0.1601 $\pm$ 0.0011 | -0.3281 $\pm$ 0.0021 | -0.1754 $\pm$ 0.0013 |
| | Norm | 0.1990 | 0.1184 | 0.1990 | 0.1476 |
| | LL | -0.2289 $\pm$ 0.0035 | -0.1691 $\pm$ 0.0012 | -0.3320 $\pm$ 0.0025 | -0.1778 $\pm$ 0.0011 |
| | FV | -0.2001 | -0.1177 | -0.1881 | -0.1433 |
| MoCo | $NC_{100}$ | 0.3586 $\pm$ 0.0049 | 0.3038 $\pm$ 0.0060 | **0.2788** $\pm$ 0.0089 | **0.2247** $\pm$ 0.0067 |
| | $Dist_1$ | -0.1344 $\pm$ 0.0046 | -0.1149 $\pm$ 0.0047 | -0.0778 $\pm$ 0.0067 | -0.0351 $\pm$ 0.0070 |
| | Norm | **0.3798** | **0.3294** | 0.2172 | 0.1659 |
| | LL | -0.1138 $\pm$ 0.0090 | -0.1032 $\pm$ 0.0093 | -0.1433 $\pm$ 0.0093 | -0.1131 $\pm$ 0.0094 |
| | FV | -0.3707 | -0.3201 | -0.2604 | -0.2145 |

Table 6: Comparison on the correlation between our neighborhood consistency ($NC_{100}$) and baseline methods in relation to performance on transfer learning tasks from `TinyImagenet` to `CIFAR-10`, `CIFAR-100`, and `STL-10`.

| Pretraining Algorithms | Method | ResNet-18 | | | | | |
| --- | --- | --- | --- | --- | --- | --- | --- |
| | | `TinyImagenet` $\rightarrow$ | | | | | |
| | | `CIFAR-10` | | `CIFAR-100` | | `STL-10` | |
| | | Entropy | Brier | Entropy | Brier | Entropy | Brier |
| SimCLR | $NC_{100}$ | $\underline{0.1729}_{\pm 0.0090}$ | $\underline{0.1292}_{\pm 0.0089}$ | $\mathbf{0.1680}_{\pm 0.0032}$ | $\mathbf{0.1343}_{\pm 0.0032}$ | $\mathbf{0.2176}_{\pm 0.0112}$ | $\mathbf{0.1513}_{\pm 0.0106}$ |
| | $Dist_1$ | $0.1311_{\pm 0.0127}$ | $0.0933_{\pm 0.0115}$ | $0.0138_{\pm 0.0071}$ | $-0.0251_{\pm 0.0049}$ | $0.1098_{\pm 0.0151}$ | $0.0520_{\pm 0.0116}$ |
| | Norm | $\mathbf{0.2124}$ | $\mathbf{0.1642}$ | $\underline{0.1220}$ | $\underline{0.0507}$ | $\underline{0.1641}$ | $\underline{0.0852}$ |
| | LL | $0.0369_{\pm 0.0063}$ | $0.0168_{\pm 0.0056}$ | $-0.0991_{\pm 0.0065}$ | $-0.1269_{\pm 0.0050}$ | $0.0205_{\pm 0.0035}$ | $-0.0180_{\pm 0.0033}$ |
| | FV | $-0.0919$ | $-0.0779$ | $-0.1660$ | $-0.1668$ | $-0.1872$ | $-0.1593$ |
| BYOL | $NC_{100}$ | $\mathbf{0.2557}_{\pm 0.0080}$ | $\mathbf{0.1431}_{\pm 0.0049}$ | $\mathbf{0.2352}_{\pm 0.0027}$ | $\mathbf{0.1525}_{\pm 0.0028}$ | $\underline{0.0374}_{\pm 0.0101}$ | $\underline{0.0232}_{\pm 0.0052}$ |
| | $Dist_1$ | $-0.1458_{\pm 0.0073}$ | $-0.0923_{\pm 0.0048}$ | $-0.2716_{\pm 0.0053}$ | $-0.1525_{\pm 0.0057}$ | $-0.2893_{\pm 0.0089}$ | $-0.2178_{\pm 0.0071}$ |
| | Norm | $\underline{0.1483}$ | $\underline{0.0749}$ | $\underline{0.1879}$ | $\underline{0.1315}$ | $\mathbf{0.1021}$ | $\mathbf{0.0772}$ |
| | LL | $-0.2106_{\pm 0.0042}$ | $-0.1248_{\pm 0.0027}$ | $-0.3418_{\pm 0.0022}$ | $-0.2006_{\pm 0.0023}$ | $-0.3319_{\pm 0.0021}$ | $-0.2472_{\pm 0.0024}$ |
| | FV | $-0.1386$ | $-0.0713$ | $-0.1874$ | $-0.1202$ | $-0.1171$ | $-0.0790$ |
| MoCo | $NC_{100}$ | $\underline{0.1880}_{\pm 0.0080}$ | $\underline{0.1313}_{\pm 0.0071}$ | $\mathbf{0.1797}_{\pm 0.0054}$ | $\mathbf{0.1446}_{\pm 0.0056}$ | $\mathbf{0.2263}_{\pm 0.0128}$ | $\mathbf{0.1463}_{\pm 0.0129}$ |
| | $Dist_1$ | $0.1213_{\pm 0.0097}$ | $0.0718_{\pm 0.0093}$ | $0.0311_{\pm 0.0060}$ | $-0.0143_{\pm 0.0057}$ | $0.0993_{\pm 0.0142}$ | $0.0223_{\pm 0.0101}$ |
| | Norm | $\mathbf{0.1930}$ | $\mathbf{0.1355}$ | $\underline{0.1321}$ | $\underline{0.0531}$ | $\underline{0.1327}$ | $\underline{0.0443}$ |
| | LL | $0.0158_{\pm 0.0087}$ | $-0.0082_{\pm 0.0074}$ | $-0.1046_{\pm 0.0048}$ | $-0.1386_{\pm 0.0040}$ | $-0.0083_{\pm 0.0074}$ | $-0.0577_{\pm 0.0067}$ |
| | FV | $-0.0793$ | $-0.0701$ | $-0.1499$ | $-0.1658$ | $-0.1615$ | $-0.1577$ |

| Pretraining Algorithms | Method | ResNet-50 | | | | | |
| --- | --- | --- | --- | --- | --- | --- | --- |
| | | `TinyImagenet` $\rightarrow$ | | | | | |
| | | `CIFAR-10` | | `CIFAR-100` | | `STL-10` | |
| | | Entropy | Brier | Entropy | Brier | Entropy | Brier |
| SimCLR | $NC_{100}$ | $\underline{0.1224}_{\pm 0.0061}$ | $\underline{0.0804}_{\pm 0.0048}$ | $\mathbf{0.1467}_{\pm 0.0075}$ | $\mathbf{0.1194}_{\pm 0.0062}$ | $\mathbf{0.1866}_{\pm 0.0062}$ | $\mathbf{0.1169}_{\pm 0.0055}$ |
| | $Dist_1$ | $0.0894_{\pm 0.0077}$ | $0.0531_{\pm 0.0081}$ | $-0.0246_{\pm 0.0072}$ | $-0.0627_{\pm 0.0066}$ | $0.0340_{\pm 0.0142}$ | $-0.0167_{\pm 0.0117}$ |
| | Norm | $\mathbf{0.1549}$ | $\mathbf{0.1098}$ | $\underline{0.0895}$ | $\underline{0.0177}$ | $\underline{0.0608}$ | $\underline{-0.0056}$ |
| | LL | $0.0225_{\pm 0.0077}$ | $0.0027_{\pm 0.0062}$ | $-0.1192_{\pm 0.0051}$ | $-0.1443_{\pm 0.0041}$ | $-0.0149_{\pm 0.0062}$ | $-0.0495_{\pm 0.0043}$ |
| | FV | $-0.1078$ | $-0.0878$ | $-0.1867$ | $-0.1840$ | $-0.2178$ | $-0.1829$ |
| BYOL | $NC_{100}$ | $\mathbf{0.2561}_{\pm 0.0066}$ | $\mathbf{0.1258}_{\pm 0.0037}$ | $\mathbf{0.2600}_{\pm 0.0028}$ | $\mathbf{0.1005}_{\pm 0.0020}$ | $\mathbf{0.1410}_{\pm 0.0054}$ | $0.0123_{\pm 0.0038}$ |
| | $Dist_1$ | $-0.1885_{\pm 0.0016}$ | $-0.1274_{\pm 0.0016}$ | $-0.3075_{\pm 0.0019}$ | $-0.1679_{\pm 0.0040}$ | $-0.2371_{\pm 0.0022}$ | $-0.1925_{\pm 0.0020}$ |
| | Norm | $\underline{0.1620}$ | $\underline{0.0918}$ | $\underline{0.1082}$ | $\underline{0.0816}$ | $\underline{0.0963}$ | $\mathbf{0.0380}$ |
| | LL | $-0.3130_{\pm 0.0040}$ | $-0.1763_{\pm 0.0029}$ | $-0.3671_{\pm 0.0031}$ | $-0.1797_{\pm 0.0019}$ | $-0.3235_{\pm 0.0094}$ | $-0.2002_{\pm 0.0048}$ |
| | FV | $-0.1668$ | $-0.0737$ | $-0.1441$ | $-0.0705$ | $-0.0584$ | $\underline{0.0205}$ |
| MoCo | $NC_{100}$ | $\underline{0.1927}_{\pm 0.0045}$ | $\underline{0.1300}_{\pm 0.0040}$ | $\mathbf{0.1962}_{\pm 0.0019}$ | $\mathbf{0.1560}_{\pm 0.0022}$ | $\underline{0.2170}_{\pm 0.0072}$ | $\underline{0.1433}_{\pm 0.0046}$ |
| | $Dist_1$ | $0.1751_{\pm 0.0081}$ | $0.1109_{\pm 0.0077}$ | $0.0759_{\pm 0.0054}$ | $0.0257_{\pm 0.0045}$ | $0.1696_{\pm 0.0106}$ | $0.0710_{\pm 0.0089}$ |
| | Norm | $\mathbf{0.2503}$ | $\mathbf{0.1803}$ | $\underline{0.1838}$ | $\underline{0.1034}$ | $\mathbf{0.2524}$ | $\mathbf{0.1524}$ |
| | LL | $0.0458_{\pm 0.0108}$ | $0.0145_{\pm 0.0098}$ | $-0.0916_{\pm 0.0060}$ | $-0.1239_{\pm 0.0055}$ | $0.0133_{\pm 0.0036}$ | $-0.0502_{\pm 0.0040}$ |
| | FV | $-0.0560$ | $-0.0400$ | $-0.1510$ | $-0.1578$ | $-0.1735$ | $-0.1577$ |

Table 7: Comparison on the correlation between our neighborhood consistency ($NC_{100}$) and baseline methods in relation to performance on transfer learning tasks from `TinyImagenet` to `CIFAR-10`, `CIFAR-100`, and `STL-10`, for the **unnormalized** representation.

| Pretraining Algorithms | Method | ResNet-18 | | | | | |
| | | TinyImagenet → | | | | | |
| | | CIFAR-10 | | CIFAR-100 | | STL-10 | |
| | | Entropy | Brier | Entropy | Brier | Entropy | Brier |
|---|---|---|---|---|---|---|---|
| SimCLR | $NC_{100}$ | $0.1392_{\pm0.0072}$ | $0.1025_{\pm0.0067}$ | $\mathbf{0.1520}_{\pm0.0052}$ | $\mathbf{0.1218}_{\pm0.0057}$ | $0.1540_{\pm0.0103}$ | $\mathbf{0.1054}_{\pm0.0080}$ |
| | $Dist_1$ | $-0.1318_{\pm0.0092}$ | $-0.1101_{\pm0.0090}$ | $-0.1067_{\pm0.0079}$ | $-0.0671_{\pm0.0053}$ | $-0.1211_{\pm0.0120}$ | $-0.0838_{\pm0.0086}$ |
| | Norm | $\mathbf{0.2124}$ | $\mathbf{0.1642}$ | $0.1220$ | $0.0507$ | $\mathbf{0.1641}$ | $0.0852$ |
| | LL | $-0.2160_{\pm0.0245}$ | $-0.1896_{\pm0.0169}$ | $-0.2046_{\pm0.0167}$ | $-0.1679_{\pm0.0121}$ | $-0.1988_{\pm0.0330}$ | $-0.1736_{\pm0.0187}$ |
| | FV | $-0.2816$ | $-0.2204$ | $-0.2308$ | $-0.1655$ | $-0.2912$ | $-0.2040$ |
| BYOL | $NC_{100}$ | $\mathbf{0.2551}_{\pm0.0078}$ | $\mathbf{0.1437}_{\pm0.0047}$ | $\mathbf{0.2317}_{\pm0.0026}$ | $\mathbf{0.1505}_{\pm0.0026}$ | $0.0354_{\pm0.0103}$ | $0.0234_{\pm0.0056}$ |
| | $Dist_1$ | $-0.1759_{\pm0.0052}$ | $-0.1076_{\pm0.0032}$ | $-0.2996_{\pm0.0045}$ | $-0.1732_{\pm0.0046}$ | $-0.3114_{\pm0.0080}$ | $-0.2346_{\pm0.0061}$ |
| | Norm | $0.1483$ | $0.0749$ | $0.1879$ | $0.1315$ | $\mathbf{0.1021}$ | $\mathbf{0.0772}$ |
| | LL | $-0.2346_{\pm0.0079}$ | $-0.1338_{\pm0.0034}$ | $-0.3422_{\pm0.0046}$ | $-0.2052_{\pm0.0028}$ | $-0.3240_{\pm0.0070}$ | $-0.2454_{\pm0.0039}$ |
| | FV | $-0.1461$ | $-0.0743$ | $-0.1886$ | $-0.1296$ | $-0.1100$ | $-0.0814$ |
| MoCo | $NC_{100}$ | $0.1728_{\pm0.0075}$ | $0.1227_{\pm0.0064}$ | $\mathbf{0.1738}_{\pm0.0042}$ | $\mathbf{0.1393}_{\pm0.0041}$ | $\mathbf{0.1886}_{\pm0.0108}$ | $\mathbf{0.1229}_{\pm0.0092}$ |
| | $Dist_1$ | $-0.0949_{\pm0.0081}$ | $-0.0767_{\pm0.0084}$ | $-0.0948_{\pm0.0071}$ | $-0.0570_{\pm0.0058}$ | $-0.0650_{\pm0.0112}$ | $-0.0440_{\pm0.0076}$ |
| | Norm | $\mathbf{0.1930}$ | $\mathbf{0.1355}$ | $0.1321$ | $0.0531$ | $0.1327$ | $0.0443$ |
| | LL | $-0.1594_{\pm0.0210}$ | $-0.1421_{\pm0.0115}$ | $-0.1662_{\pm0.0158}$ | $-0.1439_{\pm0.0061}$ | $-0.1339_{\pm0.0352}$ | $-0.1382_{\pm0.0179}$ |
| | FV | $-0.2590$ | $-0.1912$ | $-0.2404$ | $-0.1730$ | $-0.2452$ | $-0.1591$ |

| Pretraining Algorithms | Method | ResNet-50 | | | | | |
| | | TinyImagenet → | | | | | |
| | | CIFAR-10 | | CIFAR-100 | | STL-10 | |
| | | Entropy | Brier | Entropy | Brier | Entropy | Brier |
|---|---|---|---|---|---|---|---|
| SimCLR | $NC_{100}$ | $0.1061_{\pm0.0104}$ | $0.0693_{\pm0.0091}$ | $\mathbf{0.1385}_{\pm0.0102}$ | $\mathbf{0.1144}_{\pm0.0088}$ | $\mathbf{0.1216}_{\pm0.0154}$ | $\mathbf{0.0710}_{\pm0.0129}$ |
| | $Dist_1$ | $-0.0954_{\pm0.0092}$ | $-0.0787_{\pm0.0090}$ | $-0.1092_{\pm0.0099}$ | $-0.0748_{\pm0.0088}$ | $-0.0775_{\pm0.0148}$ | $-0.0544_{\pm0.0108}$ |
| | Norm | $\mathbf{0.1549}$ | $\mathbf{0.1098}$ | $0.0895$ | $0.0177$ | $0.0608$ | $-0.0056$ |
| | LL | $-0.1479_{\pm0.0121}$ | $-0.1284_{\pm0.0095}$ | $-0.1677_{\pm0.0124}$ | $-0.1412_{\pm0.0108}$ | $-0.1293_{\pm0.0209}$ | $-0.1208_{\pm0.0128}$ |
| | FV | $-0.2344$ | $-0.1760$ | $-0.2218$ | $-0.1556$ | $-0.2328$ | $-0.1520$ |
| BYOL | $NC_{100}$ | $\mathbf{0.2578}_{\pm0.0067}$ | $\mathbf{0.1273}_{\pm0.0039}$ | $\mathbf{0.2598}_{\pm0.0029}$ | $\mathbf{0.1014}_{\pm0.0021}$ | $\mathbf{0.1404}_{\pm0.0058}$ | $0.0101_{\pm0.0038}$ |
| | $Dist_1$ | $-0.2212_{\pm0.0015}$ | $-0.1375_{\pm0.0019}$ | $-0.3117_{\pm0.0011}$ | $-0.1690_{\pm0.0029}$ | $-0.2528_{\pm0.0026}$ | $-0.1894_{\pm0.0027}$ |
| | Norm | $0.1620$ | $0.0918$ | $0.1082$ | $0.0816$ | $0.0963$ | $\mathbf{0.0380}$ |
| | LL | $-0.2331_{\pm0.0036}$ | $-0.1405_{\pm0.0019}$ | $-0.3030_{\pm0.0047}$ | $-0.1634_{\pm0.0024}$ | $-0.2607_{\pm0.0043}$ | $-0.1895_{\pm0.0021}$ |
| | FV | $-0.1729$ | $-0.0944$ | $-0.1309$ | $-0.0868$ | $-0.1125$ | $-0.0368$ |
| MoCo | $NC_{100}$ | $0.2118_{\pm0.0062}$ | $0.1453_{\pm0.0052}$ | $\mathbf{0.2091}_{\pm0.0045}$ | $\mathbf{0.1575}_{\pm0.0035}$ | $0.1827_{\pm0.0088}$ | $0.1071_{\pm0.0094}$ |
| | $Dist_1$ | $-0.1440_{\pm0.0047}$ | $-0.1171_{\pm0.0049}$ | $-0.1306_{\pm0.0076}$ | $-0.0882_{\pm0.0062}$ | $-0.1507_{\pm0.0081}$ | $-0.1256_{\pm0.0052}$ |
| | Norm | $\mathbf{0.2503}$ | $\mathbf{0.1803}$ | $0.1838$ | $0.1034$ | $\mathbf{0.2524}$ | $\mathbf{0.1524}$ |
| | LL | $-0.1417_{\pm0.0191}$ | $-0.1256_{\pm0.0137}$ | $-0.1374_{\pm0.0141}$ | $-0.1093_{\pm0.0119}$ | $-0.1313_{\pm0.0257}$ | $-0.1411_{\pm0.0200}$ |
| | FV | $-0.2906$ | $-0.2087$ | $-0.2559$ | $-0.1829$ | $-0.3238$ | $-0.2290$ |

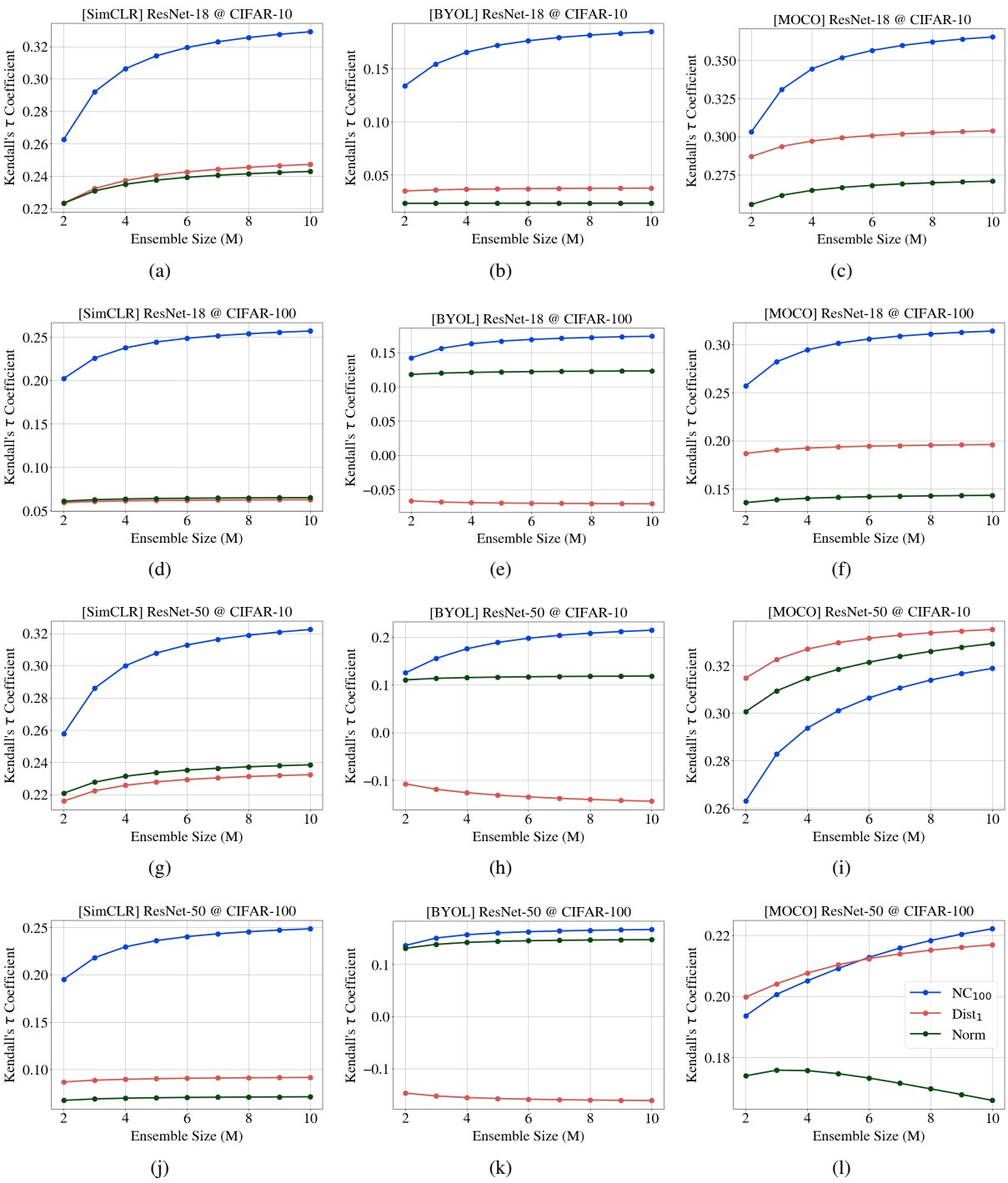

Figure 4: Ablation over the ensemble size ($M$) for $NC_k$ (ours) and baselines. Brier score is used for the downstream performance metric.

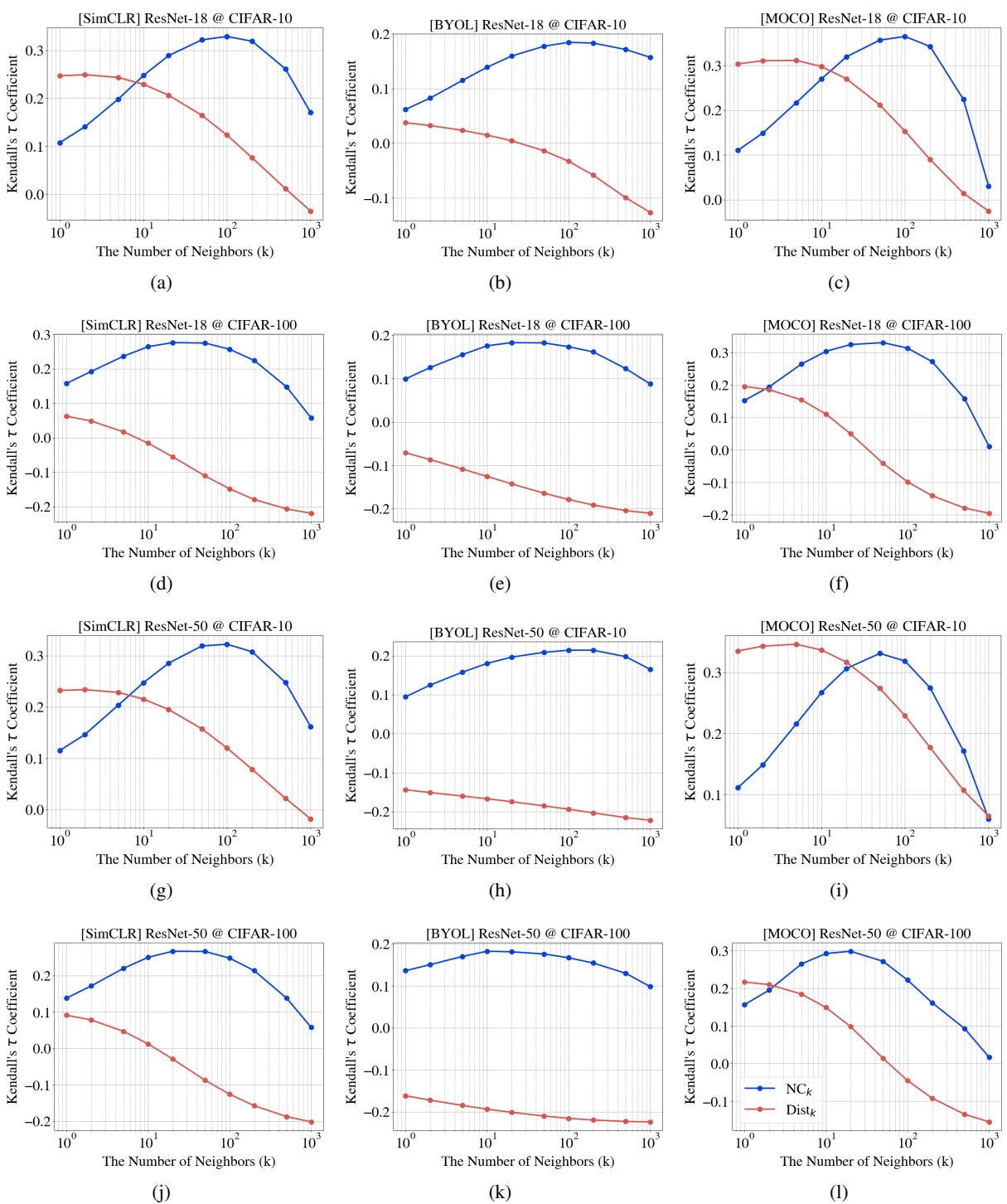

Figure 5: Ablation over the number of neighbors ($k$) for $\text{NC}_k$ (ours) and baselines. Brier score is used for the downstream performance metric.

Table 8: Comparison on the correlation between our neighborhood consistency ($NC_{100}$) and baseline methods in relation to performance on in-distribution downstream tasks for a **single embedding function**.

| Pretraining Algorithms | Method | ResNet-18 | | ResNet-50 | |
|---|---|---|---|---|---|
| | | CIFAR-10 | CIFAR-100 | CIFAR-10 | CIFAR-100 |
| SimCLR | $NC_{100}$ | **0.3069** $\pm$ 0.0056 | **0.2463** $\pm$ 0.0033 | **0.3067** $\pm$ 0.0059 | **0.2428** $\pm$ 0.0032 |
| | $Dist_1$ | 0.2412 $\pm$ 0.0020 | 0.0625 $\pm$ 0.0033 | 0.2283 $\pm$ 0.0057 | 0.0898 $\pm$ 0.0020 |
| | $Dist_1^*$ | 0.2058 $\pm$ 0.0208 | 0.0572 $\pm$ 0.0199 | 0.2048 $\pm$ 0.0290 | 0.0838 $\pm$ 0.0246 |
| | Norm | 0.2368 $\pm$ 0.0065 | 0.0628 $\pm$ 0.0048 | 0.2346 $\pm$ 0.0055 | 0.0698 $\pm$ 0.0063 |
| | Norm$^*$ | 0.2158 $\pm$ 0.0390 | 0.0621 $\pm$ 0.0352 | 0.2157 $\pm$ 0.0326 | 0.0690 $\pm$ 0.0321 |
| | LL | 0.1304 $\pm$ 0.0044 | -0.0983 $\pm$ 0.0039 | 0.1316 $\pm$ 0.0030 | -0.0907 $\pm$ 0.0033 |
| | LL$^*$ | 0.1113 $\pm$ 0.0175 | -0.0866 $\pm$ 0.0151 | 0.1188 $\pm$ 0.0234 | -0.0803 $\pm$ 0.0149 |
| | FV | -0.0431 $\pm$ 0.0080 | -0.1625 $\pm$ 0.0034 | -0.0359 $\pm$ 0.0051 | -0.1798 $\pm$ 0.0044 |
| BYOL | $NC_{100}$ | **0.1741** $\pm$ 0.0050 | **0.1685** $\pm$ 0.0026 | **0.1904** $\pm$ 0.0223 | **0.1655** $\pm$ 0.0118 |
| | $Dist_1$ | 0.0399 $\pm$ 0.0058 | -0.0623 $\pm$ 0.0024 | -0.1221 $\pm$ 0.0185 | -0.1441 $\pm$ 0.0109 |
| | $Dist_1^*$ | 0.0324 $\pm$ 0.0211 | -0.0576 $\pm$ 0.0107 | -0.0849 $\pm$ 0.0474 | -0.1270 $\pm$ 0.0141 |
| | Norm | 0.0241 $\pm$ 0.0059 | 0.1195 $\pm$ 0.0042 | 0.1100 $\pm$ 0.0292 | 0.1400 $\pm$ 0.0094 |
| | Norm$^*$ | 0.0246 $\pm$ 0.0084 | 0.1112 $\pm$ 0.0173 | 0.0895 $\pm$ 0.0373 | 0.1142 $\pm$ 0.0432 |
| | LL | -0.0157 $\pm$ 0.0060 | -0.1533 $\pm$ 0.0028 | -0.1534 $\pm$ 0.0149 | -0.1847 $\pm$ 0.0090 |
| | LL$^*$ | -0.0190 $\pm$ 0.0232 | -0.1420 $\pm$ 0.0099 | -0.1339 $\pm$ 0.0370 | -0.1681 $\pm$ 0.0116 |
| | FV | 0.0936 $\pm$ 0.0042 | -0.1236 $\pm$ 0.0056 | 0.0432 $\pm$ 0.0153 | -0.0639 $\pm$ 0.0107 |
| MoCo | $NC_{100}$ | **0.3521** $\pm$ 0.0025 | **0.3038** $\pm$ 0.0039 | 0.3137 $\pm$ 0.0039 | **0.2173** $\pm$ 0.0039 |
| | $Dist_1$ | 0.2990 $\pm$ 0.0022 | 0.1873 $\pm$ 0.0022 | **0.3296** $\pm$ 0.0033 | 0.2130 $\pm$ 0.0033 |
| | $Dist_1^*$ | 0.2763 $\pm$ 0.0280 | 0.1766 $\pm$ 0.0124 | 0.3065 $\pm$ 0.0188 | 0.1991 $\pm$ 0.0277 |
| | Norm | 0.2646 $\pm$ 0.0041 | 0.1396 $\pm$ 0.0021 | 0.3214 $\pm$ 0.0041 | 0.1616 $\pm$ 0.0058 |
| | Norm$^*$ | 0.2536 $\pm$ 0.0350 | 0.1369 $\pm$ 0.0191 | 0.2952 $\pm$ 0.0305 | 0.1750 $\pm$ 0.0421 |
| | LL | 0.1697 $\pm$ 0.0025 | -0.0710 $\pm$ 0.0030 | 0.2071 $\pm$ 0.0053 | -0.0331 $\pm$ 0.0041 |
| | LL$^*$ | 0.1545 $\pm$ 0.0261 | -0.0629 $\pm$ 0.0096 | 0.1866 $\pm$ 0.0173 | -0.0266 $\pm$ 0.0208 |
| | FV | 0.0324 $\pm$ 0.0036 | -0.1534 $\pm$ 0.0030 | 0.1101 $\pm$ 0.0042 | -0.1075 $\pm$ 0.0048 |

Table 9: Comparison on the correlation between our neighborhood consistency ($NC_{100}$) and baseline methods in relation to performance on transfer learning tasks from `TinyImagenet` to `CIFAR-10`, `CIFAR-100`, and `STL-10` for a **single embedding function**.

| Pretraining Algorithms | Method | ResNet-18 | | | ResNet-50 | | |
|---|---|---|---|---|---|---|---|
| | | TinyImagenet $\rightarrow$ | | | TinyImagenet $\rightarrow$ | | |
| | | CIFAR-10 | CIFAR-100 | STL-10 | CIFAR-10 | CIFAR-100 | STL-10 |
| SimCLR | $NC_{100}$ | $0.1256_{\pm 0.0031}$ | $\mathbf{0.1301}_{\pm 0.0023}$ | $\mathbf{0.1485}_{\pm 0.0016}$ | $0.0791_{\pm 0.0024}$ | $\mathbf{0.1171}_{\pm 0.0015}$ | $\mathbf{0.1153}_{\pm 0.0021}$ |
| | $Dist_1$ | $0.0915_{\pm 0.0036}$ | $-0.0235_{\pm 0.0018}$ | $0.0519_{\pm 0.0030}$ | $0.0531_{\pm 0.0032}$ | $-0.0611_{\pm 0.0016}$ | $-0.0161_{\pm 0.0037}$ |
| | $Dist_1^*$ | $0.0835_{\pm 0.0112}$ | $-0.0204_{\pm 0.0045}$ | $0.0490_{\pm 0.0129}$ | $0.0488_{\pm 0.0143}$ | $-0.0566_{\pm 0.0092}$ | $-0.0134_{\pm 0.0215}$ |
| | Norm | $\mathbf{0.1619}_{\pm 0.0032}$ | $\underline{0.0499}_{\pm 0.0022}$ | $0.0845_{\pm 0.0028}$ | $\mathbf{0.1093}_{\pm 0.0025}$ | $0.0179_{\pm 0.0024}$ | $-0.0051_{\pm 0.0037}$ |
| | $Norm^*$ | $\underline{0.1542}_{\pm 0.0125}$ | $0.0494_{\pm 0.0063}$ | $\underline{0.0847}_{\pm 0.0172}$ | $\underline{0.1066}_{\pm 0.0191}$ | $\underline{0.0198}_{\pm 0.0159}$ | $\underline{0.0012}_{\pm 0.0301}$ |
| | LL | $0.0179_{\pm 0.0035}$ | $-0.1221_{\pm 0.0020}$ | $-0.0165_{\pm 0.0028}$ | $0.0039_{\pm 0.0029}$ | $-0.1413_{\pm 0.0023}$ | $-0.0484_{\pm 0.0036}$ |
| | $LL^*$ | $0.0172_{\pm 0.0098}$ | $-0.1136_{\pm 0.0042}$ | $-0.0128_{\pm 0.0137}$ | $0.0032_{\pm 0.0162}$ | $-0.1328_{\pm 0.0074}$ | $-0.0438_{\pm 0.0208}$ |
| | FV | $-0.0753_{\pm 0.0026}$ | $-0.1607_{\pm 0.0024}$ | $-0.1554_{\pm 0.0019}$ | $-0.0859_{\pm 0.0044}$ | $-0.1802_{\pm 0.0023}$ | $-0.1801_{\pm 0.0048}$ |
| BYOL | $NC_{100}$ | $\mathbf{0.1362}_{\pm 0.0119}$ | $\mathbf{0.1451}_{\pm 0.0084}$ | $0.0221_{\pm 0.0080}$ | $\mathbf{0.1141}_{\pm 0.0114}$ | $\mathbf{0.0881}_{\pm 0.0078}$ | $0.0121_{\pm 0.0180}$ |
| | $Dist_1$ | $-0.0813_{\pm 0.0066}$ | $-0.1393_{\pm 0.0040}$ | $-0.2011_{\pm 0.0077}$ | $-0.1103_{\pm 0.0209}$ | $-0.1428_{\pm 0.0200}$ | $-0.1677_{\pm 0.0195}$ |
| | $Dist_1^*$ | $-0.0732_{\pm 0.0167}$ | $-0.1230_{\pm 0.0146}$ | $-0.1835_{\pm 0.0205}$ | $-0.0830_{\pm 0.0357}$ | $-0.1260_{\pm 0.0243}$ | $-0.1272_{\pm 0.0452}$ |
| | Norm | $0.0736_{\pm 0.0109}$ | $\underline{0.1254}_{\pm 0.0048}$ | $\mathbf{0.0769}_{\pm 0.0060}$ | $\underline{0.0849}_{\pm 0.0114}$ | $\underline{0.0716}_{\pm 0.0139}$ | $\mathbf{0.0392}_{\pm 0.0167}$ |
| | $Norm^*$ | $\underline{0.0738}_{\pm 0.0209}$ | $0.1191_{\pm 0.0150}$ | $\underline{0.0681}_{\pm 0.0399}$ | $0.0682_{\pm 0.0445}$ | $0.0542_{\pm 0.0512}$ | $\underline{0.0186}_{\pm 0.0637}$ |
| | LL | $-0.1131_{\pm 0.0058}$ | $-0.1857_{\pm 0.0030}$ | $-0.2295_{\pm 0.0063}$ | $-0.1555_{\pm 0.0131}$ | $-0.1537_{\pm 0.0123}$ | $-0.1753_{\pm 0.0092}$ |
| | $LL^*$ | $-0.1056_{\pm 0.0078}$ | $-0.1709_{\pm 0.0091}$ | $-0.2110_{\pm 0.0153}$ | $-0.1279_{\pm 0.0238}$ | $-0.1420_{\pm 0.0244}$ | $-0.1352_{\pm 0.0365}$ |
| | FV | $-0.0690_{\pm 0.0098}$ | $-0.1137_{\pm 0.0050}$ | $-0.0776_{\pm 0.0067}$ | $-0.0691_{\pm 0.0110}$ | $-0.0634_{\pm 0.0080}$ | $0.0152_{\pm 0.0155}$ |
| MoCo | $NC_{100}$ | $0.1290_{\pm 0.0021}$ | $\mathbf{0.1421}_{\pm 0.0012}$ | $\mathbf{0.1440}_{\pm 0.0017}$ | $0.1285_{\pm 0.0021}$ | $\mathbf{0.1541}_{\pm 0.0025}$ | $0.1419_{\pm 0.0037}$ |
| | $Dist_1$ | $0.0707_{\pm 0.0019}$ | $-0.0139_{\pm 0.0013}$ | $0.0223_{\pm 0.0017}$ | $0.1098_{\pm 0.0017}$ | $0.0253_{\pm 0.0010}$ | $0.0704_{\pm 0.0021}$ |
| | $Dist_1^*$ | $0.0667_{\pm 0.0093}$ | $-0.0109_{\pm 0.0059}$ | $0.0233_{\pm 0.0158}$ | $0.1049_{\pm 0.0106}$ | $0.0264_{\pm 0.0085}$ | $0.0688_{\pm 0.0091}$ |
| | Norm | $\mathbf{0.1337}_{\pm 0.0031}$ | $0.0520_{\pm 0.0017}$ | $0.0435_{\pm 0.0019}$ | $\mathbf{0.1785}_{\pm 0.0011}$ | $\underline{0.1019}_{\pm 0.0028}$ | $\mathbf{0.1502}_{\pm 0.0021}$ |
| | $Norm^*$ | $\underline{0.1314}_{\pm 0.0131}$ | $\underline{0.0547}_{\pm 0.0090}$ | $\underline{0.0484}_{\pm 0.0200}$ | $\underline{0.1737}_{\pm 0.0096}$ | $0.1011_{\pm 0.0126}$ | $\underline{0.1456}_{\pm 0.0149}$ |
| | LL | $-0.0075_{\pm 0.0025}$ | $-0.1360_{\pm 0.0020}$ | $-0.0565_{\pm 0.0022}$ | $0.0148_{\pm 0.0022}$ | $-0.1224_{\pm 0.0016}$ | $-0.0494_{\pm 0.0027}$ |
| | $LL^*$ | $-0.0074_{\pm 0.0109}$ | $-0.1280_{\pm 0.0038}$ | $-0.0519_{\pm 0.0147}$ | $0.0135_{\pm 0.0105}$ | $-0.1143_{\pm 0.0058}$ | $-0.0423_{\pm 0.0200}$ |
| | FV | $-0.0687_{\pm 0.0021}$ | $-0.1627_{\pm 0.0026}$ | $-0.1550_{\pm 0.0026}$ | $-0.0391_{\pm 0.0024}$ | $-0.1559_{\pm 0.0032}$ | $-0.1555_{\pm 0.0034}$ |

# D    MORE ON RELATED WORK

Self-supervised representation learning has become standard in many computer vision applications. Instead of training a neural network that takes in the raw data and outputs the target value (e.g., class label), it optimizes a neural network $h_\theta$ that maps an input $x$ into the latent vector $z \in \mathbb{R}^d$ in the $d$-dimensional representation space.

Uncertainty estimation is the process of figuring out how uncertain or reliable the learned representations of the data are. Assessing the uncertainty of the neural network's representation is a key step in making a reliable machine learning framework. This is because the uncertainty provides information about the data and how confident the model is in its modeling. There are several ways to estimate uncertainty in deep learning, such as Bayesian approaches and ensembling, in supervised learning settings where the ground truth output (e.g., label) is given. Estimating uncertainty in deep representation learning, on the other hand, is still a relatively undiscovered area of research.

## D.1    UNCERTAINTY-AWARE REPRESENTATION LEARNING

The representation model is often considered to be deterministic in recently popular frameworks [Chen et al., 2020a, He et al., 2020, Chen et al., 2020b]. To address the reliability issue of those frameworks, some recent works, including [Oh et al., 2018, Wu and Goodman, 2020], extend the prior deterministic frameworks to stochastic ones, allowing for the construction of an uncertainty-aware self-Supervised representation learning framework.

Oh et al. [2018] introduces a hedged instance embedding (HIB) that optimizes a representation network that approximates the distribution over the representation vector $p_\theta(z|x)$ under the (soft) contrastive loss [Hadsell et al., 2006] and variational information bottleneck (VIB) principle [Alemi et al., 2016, Achille and Soatto, 2017]. More specifically, HIB encoder

parameterizes the distribution as the mixture of $C$ Gaussians: $p_\theta(\boldsymbol{z}|\boldsymbol{x}) = \sum_{c=1}^{C} \mathcal{N}(\boldsymbol{z}; \mu_\theta(\boldsymbol{x}, c), \Sigma_\theta(\boldsymbol{x}, c))$. Based on the stochastic embedding, the paper proposes an uncertainty metric, called *self-mismatch* probability:

$$s_{self\_mismatch}(\boldsymbol{x}^*) \triangleq 1 - p(m|\boldsymbol{x}^*, \boldsymbol{x}^*) \tag{21}$$

where $p(m|\boldsymbol{x}_1, \boldsymbol{x}_2) \approx \int p(m|\boldsymbol{z}_1, \boldsymbol{z}_2) p_\theta(\boldsymbol{z}_1|\boldsymbol{x}_1) p_\theta(\boldsymbol{z}_2|\boldsymbol{x}_2) d\boldsymbol{z}_1 d\boldsymbol{z}_2$ and $p(m|\boldsymbol{z}_1, \boldsymbol{z}_2) \triangleq \sigma(-a\|\boldsymbol{z}_1 - \boldsymbol{z}_2\|_2 + b)$ based on their contrastive learning method. Self-mismatch probability can be interpreted as an expectation of the distance between two points randomly sampled from the output distribution. In other words, this uncertainty metric is based on the idea that an input with a large aleatoric uncertainty will span a wider region, resulting in a smaller $p(m|x, x)$.

In [Wu and Goodman, 2020], a similar extension is also proposed. The paper introduces a distribution encoder that outputs the representation of Gaussian distribution with diagonal covariance matrix $\Sigma_\theta(x)$ and extends the normalized temperature-scaled cross-entropy loss (NT-Xent) [Chen et al., 2020a] to distribution-level contrastive objective. The norm of the covariance matrix determined by the distribution encoder is used to assess the reliability of a given input:

$$s_{var}(x^*) \triangleq ||\Sigma_\theta(x^*)|| \tag{22}$$

Despite the benefits of stochastic representation, there are still some shortcomings. One limitation would be that it requires re-training. Large models that have gotten a lot of attention lately are usually trained on a lot of data and are getting bigger, which means they take more time and computing power to train. As a result, it may not always be practical or feasible for users to re-train a model. Additionally, new training schemes can impose unexpected inductive bias to algorithms that are already working well. For example, most probability-based methods are based on standard distributions like the Gaussian or a mixture of them. However, these assumptions may reduce the effectiveness of the model or slow down the training procedure.

## D.2  NOVELTY DETECTION IN REPRESENTATION SPACE

There are several studies that introduce ways to detect out-of-distribution (OOD) samples by determining the novelty of the data representation from a deterministic model [Lee et al., 2018, van Amersfoort et al., 2020, Tack et al., 2020, Mirzaei et al., 2022]. Although the specific details of each technique vary, this study observed that these methods commonly use the relative distance information of the query data point's representation vector to other reference points:

$$s_d(\boldsymbol{x}^*) \triangleq \mathsf{Dist}_k\Big(\big\{\mathsf{dist}\big(h(\boldsymbol{x}^*), h(\boldsymbol{x})\big) \mid \boldsymbol{x} \in \boldsymbol{X}_{\mathrm{ref}}\big\}\Big) \tag{23}$$

where $\mathsf{Dist}_k$ outputs an average of the $k$ smallest relative distances in the representation space between the query $\boldsymbol{x}^*$ and reference points $\boldsymbol{X}_{\mathrm{ref}}$, measured by the distance metric dist.

As shown in the table, some works are designed for supervised learning schemes that require instance-specific training labels. Lee et al. [2018] and van Amersfoort et al. [2020] construct reference points by empirical class means:

$$\left\{\hat{\boldsymbol{\mu}}_c = \frac{1}{N_c} \sum_{\boldsymbol{x}_i \in \boldsymbol{X}_{\mathrm{ref}}} h_\theta(\boldsymbol{x}_i) \mathbb{1}(y_i = c)\right\}_{c=1}^{C} \tag{24}$$

where $\hat{\boldsymbol{\mu}}_c$ is a centroid of training representations of which the label $y_i$ is equal to $c$, $N_c$ is the number of training instances belonging to the label $c$, and $C$ is the number of classes. Then, Lee et al. [2018] defines an uncertainty score as a minimum Mahalanobis distance to each of the centroids using a tied empirical covariance matrix, whereas van Amersfoort et al. [2020] calculates a distance using a Radial Basis Function (RBF) kernel. These approaches can also be viewed as estimating the probability density of the representation to each class.

Nevertheless, the aforementioned methods necessitate training labels, which is not always feasible. In addition, evaluating the uncertainty based on the training labels does not guarantee the method's efficacy, as downstream tasks frequently use distinct labeling schemes. In order to estimate uncertainty in the absence of class label information, Tack et al. [2020] measures the minimum distance between the query instance and all training instances in the representation space. Tack et al. [2020] additionally suggest to ensemble the uncertainty score with various transformations (i.e., augmentation) $\mathcal{T}$: $s_{d\text{-}ens}(\boldsymbol{x}^*) = \frac{1}{|\mathcal{T}|} \sum_{t \sim \mathcal{T}} s_d(t(\boldsymbol{x}^*))$. Meanwhile, Mirzaei et al. [2022] averages the distance to $k$-nearest instances rather than taking the closest one to improve the effectiveness.

The main limitation of the above two approaches is the lack of theoretical justification for the proposed metric, as it is founded upon heuristic rules. For example, as demonstrated by our empirical studies in Section C.1, a wrong selection of the

distance metric can lead to contradictory results. In addition, as depicted in Figure 5, using a larger $k$ in those schemes does not necessarily ensure its effectiveness. Considering that the primary goal of these studies is to deploy foundation models in safety-critical settings, establishing a robust reliability measure and analyzing its theoretical validity is vital.