# OpenReview forum: "Quantifying Representation Reliability in Self-Supervised Learning Models"
_auai.org/UAI/2024/Conference — UAI 2024 poster_

### Official Review · Reviewer_JgHt · 2024-03-05

**Q2-1 Originality-Novelty:** 3
**Q2-2 Correctness-Technical Quality:** 3
**Q2-5 Clarity Of Writing:** 3

**Q1 Summary And Contributions:**

Self-supervised learning is becoming ever more popular and assessing the quality of pretrained models in a task-agnostic fashion is therefore a highly relevant problem. Here, the authors begin by proposing a formal definition of representation reliability over a set of downstream tasks. Given this definition, they briefly review why approaches from the supervised setting aren't directly applicable to the setting without labels, motivating the remainder of this work. The authors establish a theoretical bound on the reliability of a test point given a consistent anchor point, whose reliability is known. They then propose a practical measure for the reliability of representations without access to such an anchor point and demonstrate its merit in experiments.

Overall, this work proposes an elegant approach to the problem of unsupervised model selection for self-supervised pretrained model, that could be practically useful for a wide audience.

**Q2-3 Extent To Which Claims Are Supported By Evidence:**

3: Good: the main claims are supported by convincing evidence (in the form of adequate experimental evaluation, proofs, (pseudo-)code, references, assumptions).

**Q2-4 Reproducibility:**

3: Good: key resources (e.g. proofs, code, data) are available and key details (e.g. proofs, experimental setup) are sufficiently well-described for competent researchers to confidently reproduce the main results.

**Q3 Main Strengths:**

- Addresses a highly relevant problem setting. There is a large body of work on identifiable representation learning where the ground truth is known and the evaluation of the learned representation is fairly straightforward. This complementary view of a similar problem where there is no equivalent notion of ground truth representation is insightful, both to the SSL community and for representation learning in general.

- The proposed solution to the considered problem of unsupervised representation evaluation is simple, and as the experiments demonstrate, effective. The authors' approach is clearly motivated and the additional theoretical segment is a welcome addition.

- This work is clearly and well-motivated (since the supervised approach doesn't work here) and well written, making it easy to follow.

**Q4 Main Weakness:**

- The experiment for ranking different pre-trained models is only partially convincing and I believe this could be done in a more straightforward way. This experiment reflects the primary motivation of the whole work: given different models, select the one with most reliable representation for downstream tasks. Could you show the different values of Reli for each model/dataset and demonstrate that your metric ranks these models accordingly, or at least selects the best one? To me, this is somewhat obscured by reporting the averaged correlation, and I am wondering if it truly shows the some thing. Given a more straightforward presentation of this experiment, I am willing to increase my score.
- Some of the experimental results could be discussed in greater detail, which I will comment on further in Q5.

**Q5 Detailed Comments To The Authors:**

- A priori, it was not clear to me why it is valid to consider the average over an ensemble of embedding functions in the proposed metric, if the goal is to evaluate the reliability of a single representation. After reading the paper, this is clear, but it would help to state near the beginning where the metric is introduced where this ensemble comes from and how its elements relate to each other.
- The tradeoff gauged by $k$ between including many consistent neighbors and increasing their overall reliability is not clear to me. Given the results discussed in Sec. 4.4, there clearly is a tradeoff, however I don't see why this is what has to be traded off. You claim the choosing a small value of $k$ leads to the reliability of consistent neighbors increasing. Can you explain why? By your original argument to include more test points, therefore increasing the likelihood of reliable and consistent neighboring points, couldn't one also argue that given more points (higher $k$) we increase the likelihood of consistent and reliable neighbors?
- Why is the Norm baseline so good in the transfer learning experiment?
- Why is the FV baseline so good in the model ranking experiment, especially if it shouldn't do well at all?
- In the model ranking experiment, you state that your proposed method consistently exhibits a favorable performance, which I think is an overstatement, given that FV performs better in all ResNet-18 settings.
- Your theoretical bound relies on the prediction heads being Lipschitz (which is well-justified). Please state this somewhere before the theorem to make this assumption explicit.
- The bridge from the theoretical result (Thm. 2) to the practical metric could be a bit more detailed. The results are overall positive, indicating that the proposed approach does what it is intended to in practice, but the argument that by considering more points, we are more likely to include consistent and reliable ones (while I would in general agree), doesn't necessarily seem like the only explanation. If there are no reliable samples at all, including more might not actually help. Would the metric then still be able to tell us that the representation just isn't reliable?

Minor comments:

- Fig. 1 is not directly clear without reading the rest of the paper. Perhaps you could give a slightly more detailed explanation in the caption. Specifically the distances illustrated in the $\mathcal{Y}_t$ space I find somewhat confusing.
- It would be nice to empirically see how tight the theoretical bound you propose is. If you have some results in this direction I would be curious to know, but I am not asking explicitly for more experiments if not.
- I would propose to turn Lemma 2 into a corallary or just a comment after Lemma 1, since it follows so directly

**Q9 Complying With Reviewing Instructions:**

Yes

---

> ### Author Rebuttal · Authors · 2024-04-05
>
> We thank the reviewer for carefully reading our paper and for the thoughtful comments.
>
> ---
>
> **Q1. Ranking Experiments**
>
> A1. Absolutely, this is a great suggestion! We present tables that include the values of representation reliability and various metrics for each model/dataset: [https://anonymous.4open.science/r/uai-368-610B](https://anonymous.4open.science/r/uai-368-610B).
>
> From these Tables, we observe that:
>
> i) Overall, SimCLR and MoCo achieve higher representation reliability than BYOL. The reliability of SimCLR and MoCo is comparable with SimCLR showing slightly better results.
>
> ii) Our NC consistently assigns lower scores to BYOL compared to SimCLR and MoCo across all different settings. This observation demonstrates the effectiveness of NC in identifying unreliable embedding functions.
>
> iii) FV effectively evaluates the reliability of embedding functions for ResNet-18. However, when considering ResNet-50, FV does not clearly separate BYOL from SimCLR and MoCo
>
> iv) Other baselines struggle to rank the reliability of embedding functions.
>
> ---
> **Q2. Presentation about problem statement**
>
> A2. Thank you for highlighting this matter! To address your concerns, we will add that “the ensembles are generated by using the same algorithm, architecture, and pre-training dataset but with different random seeds” and that “these embedding functions possess similar inductive biases and generate representation spaces with comparable reliability”. We will make sure to include the above discussions in the final version.
>
> ---
> **Q3. Tradeoff gauged by k**
>
> A3. Thank you for raising this important point! In response to your comments, we will revise our statements as follows:
> The value of $k$ involves a trade-off between incorporating closer neighbors and *increasing the chance of including a more reliable neighbor*. As Theorem 2 outlines, our algorithm relies on two assumptions: (1) the presence of neighbors to the test point, and (2) the reliability of the neighboring point. If we choose a large $k$, the first assumption can be violated. On the other hand, since the upper bound holds using any one of the neighbors, the reliability of the test point is bounded by the one with the smallest $\sigma_{r,t}$. Consequently, selecting a small value of $k$ may compromise the reliability of the selected neighboring points.
>
> ---
> **Q4. Analyzing the performance of Norm**
>
> A4. Indeed, this is a great observation. Several factors, including properties of embedding functions and characteristics of test points, play a crucial role in determining the reliability of a test point's representation. Our hypothesis is that in the context of transfer learning, out-of-distribution (OOD) test points are more likely to be unreliable. The baseline we've selected, Norm, is good at identifying OOD samples [Tack et al., 2020], thereby enabling it to capture representation reliability to some extent. However, in the case of in-distribution tasks (see Table 1), Norm demonstrates a lower correlation with representation reliability.
>
> ---
> **Q5. Analyzing the performance of FV**
>
> A5. FV fails to capture the reliability of each instance’s representation since it does not align different representation spaces before comparing them (Table 1, 2, and Theorem 1). However, when ranking different embedding functions, the degrees of misalignment between their ensembles are roughly the same, as these embedding functions share the same architecture. Consequently, this error term cancels out when ranking these embedding functions. This explains why FV demonstrates strong performance only in the last experiment. In other words, quantifying instance-level reliability is more challenging than the aggregate one as in Table 3. We will add this discussion to the final version.
>
> ---
>
> **Q6. Rephrasing the statement**
>
> A6. In response to your concerns, we will replace the statement as: “Our proposed method demonstrates the second-best performance for the ResNet-18 architecture and the best performance for the ResNet-50 architecture, compared to baseline methods.”
>
> ---
> **Q7. Lipschitz assumption in Theorem 2**
>
> A7. We will mention the Lipschitzness assumption before the theorem in the revised version.
>
> ---
> **Q8.  Further explanation on theorem and algorithm**
>
> A8. Theorem 2 provides a sufficient condition for identifying reliable test points. Thus, even if the bound yields a large value, it does not necessarily imply the unreliability of the test point. This scenario could occur, as you pointed out, when reference points are unreliable. Empirically, we observed that even with unreliable reference points, NC still exhibits a weak positive correlation with the representation reliability. In contrast, baselines, such as Dist_1, can even show a negative correlation. We will include these discussions in the Future work section.
>
> ---
> **Q9. Minor comments**
>
> A9. Thank you for your careful reading of our manuscript. We will update the manuscript accordingly.

---

### Official Review · Reviewer_SanU · 2024-03-09

**Q2-1 Originality-Novelty:** 2
**Q2-2 Correctness-Technical Quality:** 3
**Q2-5 Clarity Of Writing:** 4

**Q1 Summary And Contributions:**

This paper presents a notion of neighborhood consistency to quantify the representation reliability in self-supervised learning. The authors first constructed an example together with its proof showing that prior uncertainty quantification methods for supervised learning failed to work for self-supervised learning. Then, the authors introduced neighborhood consistency to facilitate the alignment of representation spaces. Extensive numerical experiments were conducted to validate the superiority of the proposed method w.r.t. representation reliability.

**Q2-3 Extent To Which Claims Are Supported By Evidence:**

2: Fair: the main claims are somewhat supported by evidence (but the experimental evaluation may be weak, or does not match entirely with the claims, important baselines may be missing, proofs contain important ideas but lack rigor, algorithmic details are only discussed superficially, references are imprecise, assumptions are not sufficiently motivated or explicated, etc.).

**Q2-4 Reproducibility:**

3: Good: key resources (e.g. proofs, code, data) are available and key details (e.g. proofs, experimental setup) are sufficiently well-described for competent researchers to confidently reproduce the main results.

**Q3 Main Strengths:**

- The reviewer finds the notion of neighborhood consistency novel and interesting. Importantly, the method does not rely on ex ante information of downstream tasks, and could be implementable in practice.
- The theoretical results are neat, which well motivate the design of quantification method for self-supervised learning.
- The experimental results look convincing. The proposed method was tested using various datasets/models/algorithms, and outperformed the baselines in most cases.
- The reviewer finds the paper to be very well written and easy to read.

**Q4 Main Weakness:**

- Although the experiments showed that neighborhood consistency (NC) outperformed baselines such as Dist_k, Norm, and FV, the reviewer finds the intuitions behind these metrics rather similar, i.e., they are based on various similarity measures of close reference points. In other words, it is unclear why NC has performance superiority. It would be nice if the authors could add some theoretical analysis or at least, construct examples to show why.
- All the results/comparisons are measured using the Kendall's coefficient. What happens if a different correlation metric is used? Also, the reviewer thinks that the ultimate goal is to obtain good performance for downstream tasks, and it would be nice to show the accuracy results for all the methods.
- The current results are based on ResNet-18 and ResNet-50. It would be nice if the authors try more recent models for self-supervised learning such as vision transformer.
- The authors claimed that NC has a ``strong'' correlation with representation reliability. However, most values in Tables 1-3 do not exceed 0.4, and in particular, 0.2 in Table 2. The reviewer is not sure whether this should be considered a strong positive correlation, or a weak-to-moderate correlation.
- In Table 3, NC seems to lag behind FV by a large margin using ResNet-18. An explanation is needed here.

**Q5 Detailed Comments To The Authors:**

See above

**Q9 Complying With Reviewing Instructions:**

Yes

---

> ### Author Rebuttal · Authors · 2024-04-05
>
> We thank the reviewer for the thoughtful comments and for appreciating the novelty of the work!
>
> ---
> **Q1. Similar intuitions behind selected baselines and why NC has performance superiority.**
>
> A1. Thank you for highlighting this matter! To clarify, the baselines we chose consist of standard metrics derived from either OOD detection or UQ in supervised learning. In OOD detection (e.g., Dist_k), closeness to in-distribution samples is crucial for identifying OOD instances; in UQ for supervised learning (e.g., FV), the consistency of neural network outputs (e.g., predictive scores) is important for assessing uncertainty. This is why these baselines share similar intuition.
>
> Next, we provide an example to illustrate why NC often has performance superiority: imagine a test point (a fox image) located near a reference point A (a dog image) and is far from another point B (a cat image) in a representation space, while the opposite holds true in another representation space. In this case, NC would indicate “unreliable” for the test point due to this inconsistency. However, baselines may suggest “reliable” since $\mathsf{Norm}$ and $\mathsf{Dist_k}$ rely on the relative distance of the test point to the origin or to the closest reference point, respectively. None of the baselines align different representation spaces before computing these relative distances. Similarly, FV fails to capture the reliability of each instance’s representation since it does not align different representation spaces before comparing them (Table 1, 2, and Theorem 1). Hence, they fail to ensure that corresponding regions in both representations carry similar semantic meanings.
>
> We will add this discussion to the final version.
>
> ---
> **Q2. Further experimental results with different metrics**
>
> A2. Indeed, this is a great suggestion! Empirically, we observed that the results are quite similar across different correlation metrics including Pearson, Kendall, Spearson’s correlation. We selected Kendall's correlation for our evaluation metric due to its robustness in non-linear relationships. We reproduce our experimental result with Pearson and Spearman's correlation to the representation reliability measured by Brier score, for in-distribution setting using SimCLR, CIFAR-10, and ResNet-18. Furthermore, we add the experimental result using accuracy as a performance metric to address your concern. The result is in the following Table. As illustrated, our observation is consistent.
>
> _Table 1: Reproduced results with different metrics_
>
> | |Brier-Spearman| Brier-Pearson| Accuracy-Kendall|
> |:----|:----|:----|:----|
> |NC_100|0.4773|0.2602|0.1972| |
> |Dist_1|0.3659|0.2048|0.1513| |
> |Norm|0.3588|0.1702|0.1301| |
> |LL|0.1944|0.0998|0.0579| |
> |FV|-0.0700|-0.0489|-0.0379| |
>
>
>
> ---
> **Q3. Extending the experiment to vision transformers (ViTs)**
>
> A3. Since we used the original codes from each cited reference, adapting our framework to include vision transformers would require further implementation efforts and additional computational resources. The one-week rebuttal period was too short to carry out these experiments but we will make every effort to incorporate these results into the final manuscript.
>
> ---
>
> **Q4. Rephrasing the statement**
>
> A4.  We completely agree. In response to your concern, we will rephrase this statement and make it more precise and consistent with our claim in Section 4.6: “while the baselines may occasionally surpass NC, their performance fluctuates significantly across different settings and sometimes becomes even negative, introducing a risk when used to assess reliability in safety-critical settings. In contrast, our method exhibits a positive correlation with the representation reliability across all different settings.”
>
> ---
>
> **Q5. In Table 3,  NC seems to lag behind FV by a large margin using ResNet-18**
>
> A5. Thank you for raising this point. While NC underperforms compared to FV in ranking experiments using ResNet-18, it's important to emphasize NC's consistency in prediction. Table 3 illustrates that NC maintains a stable positive correlation for both ResNet-18 and ResNet-50, contrasting with FV's fluctuating performance across different architectures. Moreover, Theorem 1 as well as the empirical findings in Tables 1 and 2 suggest that FV's approach of directly assessing variance in the representation space does not guarantee the reliability of the representation. Nevertheless, these are certainly interesting observations that open avenues for further investigation.
>
> It is worth noting that quantifying instance-level reliability (as in Tables 1 and 2) is more challenging than the aggregate one as in Table 3. When ranking different embedding functions, the degrees of misalignment between their ensembles are roughly the same, as these embedding functions share the same architecture. Consequently, this error term cancels out when ranking these embedding functions.

---

### Official Review · Reviewer_FNYk · 2024-03-15

**Q2-1 Originality-Novelty:** 3
**Q2-2 Correctness-Technical Quality:** 2
**Q2-5 Clarity Of Writing:** 4

**Q1 Summary And Contributions:**

This paper focused on quantifying the reliability of learned representations without any information on downstream tasks. The authors first prove a theorem that gives a sufficient condition to guarantee the bound of reliability of a test data point. Inspired by this theorem, a practical algorithm to compute the reliability score is introduced. Empirical studies show the proposed method can capture reliability better than the baselines.

**Q2-3 Extent To Which Claims Are Supported By Evidence:**

3: Good: the main claims are supported by convincing evidence (in the form of adequate experimental evaluation, proofs, (pseudo-)code, references, assumptions).

**Q2-4 Reproducibility:**

3: Good: key resources (e.g. proofs, code, data) are available and key details (e.g. proofs, experimental setup) are sufficiently well-described for competent researchers to confidently reproduce the main results.

**Q3 Main Strengths:**

[Originality/novelty] This paper's main novelty is that it studies how to quantify reliability when no downstream tasks are provided. In contrast, previous methods rely on information from output space. It also probably shows that previous techniques working on output space cannot be adapted to representation spaces directly. Based on this motivation, it provides a novel method inspired by a theoretical result, which gives a sufficient condition to guarantee the bound of downstream prediction.

[Reproducibility] The proof details are presented, and key details in the experiments are provided.

[Clarity of writing] The whole presentation is well-structured, and most of the necessary details to follow the content are provided. Also, the intuition behind the method is well-presented.

**Q4 Main Weakness:**

[Extent to which claims are supported by evidence]

1. The authors find a sufficient condition to control the variance of downstream prediction and then proposed a metric to measure the reliability of representations. However, since it is just a sufficient condition, it is possible that representation does not satisfy this condition but is still reliable for downstream. Therefore the metric cannot reflect the reliability correctly.

2. The gap between the practical algorithm proposed in this paper and the theoretical results not being analyzed, where the algorithm is a kind of approximation of the theorem.

**Q5 Detailed Comments To The Authors:**

My main considerations are the two points in Q4. If there is any misunderstanding in my interpretation, please correct me. I would greatly appreciate any discussion or clarification on the questions I have raised.

**Q9 Complying With Reviewing Instructions:**

Yes

---

> ### Author Rebuttal · Authors · 2024-04-05
>
> We thank the reviewer for the thoughtful review and for acknowledging the merits of our work.
>
> ---
> **Q1. Comment on Theorem 2 –  sufficient condition for representation reliability**
>
> A1. You are absolutely correct: Theorem 2 is a sufficient condition for detecting reliable test points. It serves as an inspiration for our algorithm: if we can find a reference point consistently close to the test point, it can serve as an anchor point that facilitates the alignment and comparison of representation spaces with distinct semantic meanings. Exploring necessary conditions is certainly an interesting avenue for future research.
>
> ---
> **Q2. Comment on algorithm – analysis on the gap between the theorem and algorithm**
>
> A2. Thank you for bringing up this crucial point. Theorem 2 only provides an upper bound for estimating representation reliability so we completely agree that there could be a gap between this theorem and the algorithm. Nevertheless, we strive to be comprehensive in our experiments: we cover a wide range of scenarios (e.g., transfer learning for OOD tasks), pre-training models, and datasets and as shown our method (NC) consistently captures representation reliability.
>
> We will incorporate these discussions into both Sections 3 and 5 (limitations) in the revised paper.

---

### Official Review · Reviewer_EhmL · 2024-04-01

**Q2-1 Originality-Novelty:** 2
**Q2-2 Correctness-Technical Quality:** 2
**Q2-5 Clarity Of Writing:** 3

**Q1 Summary And Contributions:**

This paper introduces a method to quantify the reliability of representations learned by pretrained models. It then presents a novel approach to estimate this reliability without requiring prior knowledge of the downstream tasks.

**Q2-3 Extent To Which Claims Are Supported By Evidence:**

2: Fair: the main claims are somewhat supported by evidence (but the experimental evaluation may be weak, or does not match entirely with the claims, important baselines may be missing, proofs contain important ideas but lack rigor, algorithmic details are only discussed superficially, references are imprecise, assumptions are not sufficiently motivated or explicated, etc.).

**Q2-4 Reproducibility:**

3: Good: key resources (e.g. proofs, code, data) are available and key details (e.g. proofs, experimental setup) are sufficiently well-described for competent researchers to confidently reproduce the main results.

**Q3 Main Strengths:**

1. The paper's concept is intriguing and appears to hold significant practical value, particularly regarding foundation models.
2. It examines the drawbacks of current prediction-based uncertainty quantification methods and outlines the rationale for developing a method to measure representation uncertainty.

**Q4 Main Weakness:**

1. The concept appears to be derived from an insight that lacks theoretical backing. This will be further elaborated upon in the detailed comments section.
2. The motivation behind developing uncertainty quantification (UQ) in the latent space and its comparison with UQ in supervised learning appears confusing. Additional clarity is needed, as outlined in the detailed comments section.
3. The experiments lack clarity and thorough analysis, requiring further elaboration.

**Q5 Detailed Comments To The Authors:**

1. The proposed algorithm seems to build on the insight that "a test point with more consistent neighbors is more likely to have a reliable and consistent neighbor." This claim lacks theorem results. I carefully read Theorem 1 and Theorem 2. Even though I find Theorem 1 obvious and meaningless, I agree with the results presented in those theorems and their intuition, especially for Theorem 2. The reliability of a test point corresponds to both the closeness of this test point to a reliable anchor $\varepsilon_{nb}$ and the reliability of this anchor $\sigma_{r,t}$. I understand that without prior knowledge of downstream tasks, the challenges of quantifying this representation reliability significantly increase. However, I do think there are certain assumptions for this insight to hold, for example, under an in-distribution setting. I think at least those assumptions should be explicitly mentioned.

2. I am confused by the authors' claims in the paragraph on "novelty detection and representation reliability." From my understanding, the paper aims to perform UQ for self-supervised learning in the latent space. However, it mentions OOD detection but claims that representation reliability and OOD detections are different concepts. Are the authors trying to say that this representation reliability quantification method is not applicable for OOD detection?

3. [minor]I am wondering what would happen if the authors compared with the UQ methods derived for supervised learning empirically.

5. For the pre-training stage, what is the training objective? Is it reconstruction loss over input? What are the differences between SimCLR, BYOL, and MOCO? Are there differences in architecture, training procedures, or training objectives?

6. Does the baseline Feature Variance (FV) require the class label during training?

7. In Table 2 and Table 3, do the bolded results indicate the optimal results? If so, then NC$_{100}$ does not achieve the optimal results in all settings. Why do the authors claim that in Table 2, NC demonstrates superior performance over different pre-training algorithms, datasets, and models? I feel the performance of the proposed method is worse in OOD settings (Table 2) than in in-distribution settings (Table 1). Is there any analysis regarding this? Additionally, FV performs extremely well on OOD settings with ResNet-18, much better than the proposed method. Is there any reason for this?

**Q9 Complying With Reviewing Instructions:**

Yes

---

> ### Author Rebuttal · Authors · 2024-04-05
>
> We thank the reviewer for their time and useful feedback.
>
> ---
> **Q1. Assumptions for the insight to hold**
>
> A1. Thanks for bringing up this matter. To ensure the insight holds, two assumptions must be met: (1) the test point is either in-distribution or has neighboring reference points that are close to it, (2) there are a sufficient number of reliable reference points available.
>
> Specifically, suppose the distance from the test point to its neighboring points ($\epsilon_{nb}$) remains small, which should hold true if the first assumption is met. Since our upper bound holds using any one of these neighboring points, the reliability of the test point is bounded by using the neighboring point with the smallest $\sigma_{r,t}$. Therefore, if the NC score is high – which means that there are more consistent neighbor*s* – it's more likely that at least one of them is reliable. To ensure the second assumption is satisfied, we select a subset of pre-training points to serve as reference points. We will make these assumptions and justifications explicit in the revised version.
>
> Finally, to support these claims, we conduct comprehensive experiments, i.e., we cover a wide range of scenarios (e.g., transfer learning tasks), pre-training models, and datasets, and as shown, our method (NC) consistently and effectively captures representation reliability.
>
> ---
> **Q2. OOD detection and representation reliability**
>
> A2. Thank you for raising this crucial point! What we meant to say is that representation reliability can vary even within in-distribution samples; hence, OOD detection alone may not suffice to detect unreliable test points. To address your concern, we will revise that paragraph to “OOD detection and representation reliability are different concepts. The former identifies whether a test point belongs to the same distribution as the (pre-)training data, while the latter evaluates the possibility that a test point can receive accurate predictions when the self-supervised learning model is adapted to various downstream tasks”.
>
> ---
> **Q3. Compare with the UQ methods derived for supervised learning**
>
> A3. Indeed, this is a great point! Feature Variance (FV), a baseline we have selected, is derived from UQ in supervised learning. Specifically, several studies, including [Lakshminarayanan et al. (2017)], [Ritter et al. (2018)], have suggested using the variance of neural networks’ predicted score as a measure of epistemic uncertainty. Here we adapt their method to analyze embedding/latent spaces. We will emphasize this in the revised version.
>
> ---
> **Q4. Pre-training objectives of SimCLR, BYOL, and MOCO**
>
> A4. SimCLR, BYOL, and MoCo have different pre-training procedures but share the same architecture and datasets. Specifically, SimCLR and MoCo both use infoNCE loss function, but MoCo uses a dynamic dictionary with a queue and a moving-averaged encoder to maintain consistency over time, whereas SimCLR relies on in-batch negatives. In contrast, BYOL eliminates the need for negative pairs by learning representations through a self-distillation approach. We will add these details to the final version.
>
> ---
> **Q5. Does FV require the class label during training?**
>
> A5. To clarify, our experiments aim to validate if we can estimate representation reliability without knowing downstream tasks a priori. Hence, neither the baseline methods nor our NC require downstream class labels or any additional training – otherwise, one can compute representation reliability directly from its definition.
>
> ---
>
> **Q6. Analysis on empirical performance**
>
> A6. Yes, the best and second-best results are highlighted in bold and underlined, respectively. As stated in Section 4.6, while the baselines occasionally surpass NC, their performance fluctuates significantly across different settings and sometimes becomes even negative, introducing a risk when used to assess reliability in safety-critical settings. In contrast, we would like to highlight that our NC consistently shows robust performance. To address your concerns, we will revise the text related to those Tables accordingly.
>
> FV fails to capture the reliability of each instance’s representation since it does not align different representation spaces before comparing them (Table 1, 2, and Theorem 1). However, when ranking different embedding functions, the degrees of misalignment between their ensembles are roughly the same, as these embedding functions share the same architecture. Consequently, this error term cancels out when ranking these embedding functions. This explains why FV demonstrates strong performance only in the last experiment. In other words, quantifying instance-level reliability (as in Tables 1 and 2) is more challenging than the aggregate one as in Table 3. We will add this discussion to the final version.

---

### Meta-Review · Area_Chair_6Uni · 2024-04-15

This paper presents a method based on neighborhood consistency to quantify representation reliability in self-supervised learning.

In the reviews, reviewers acknowledged the strong motivation, novel idea, empirical performance and writing quality. After the rebuttal, reviewers confirmed that major concerns have been addressed, and all reviewers rate positive on this paper.